# Reciprocally inhibitory circuits operating with distinct mechanisms are differently robust to perturbation and modulation

Ekaterina Morozova[1], Peter Newstein[1,2], Eve Marder[1]*

[1]Volen Center and Department of Biology, Brandeis University, Waltham, United States; [2]Biology Department, University of Oregon, Eugene, United States

**Abstract** Reciprocal inhibition is a building block in many sensory and motor circuits. We studied the features that underly robustness in reciprocally inhibitory two neuron circuits. We used the dynamic clamp to create reciprocally inhibitory circuits from pharmacologically isolated neurons of the crab stomatogastric ganglion by injecting artificial graded synaptic ($I_{Syn}$) and hyperpolarization-activated inward ($I_H$) currents. There is a continuum of mechanisms in circuits that generate antiphase oscillations, with 'release' and 'escape' mechanisms at the extremes, and mixed mode oscillations between these extremes. In release, the active neuron primarily controls the off/on transitions. In escape, the inhibited neuron controls the transitions. We characterized the robustness of escape and release circuits to alterations in circuit parameters, temperature, and neuromodulation. We found that escape circuits rely on tight correlations between synaptic and H conductances to generate bursting but are resilient to temperature increase. Release circuits are robust to variations in synaptic and H conductances but fragile to temperature increase. The modulatory current ($I_{MI}$) restores oscillations in release circuits but has little effect in escape circuits. Perturbations can alter the balance of escape and release mechanisms and can create mixed mode oscillations. We conclude that the same perturbation can have dramatically different effects depending on the circuits' mechanism of operation that may not be observable from basal circuit activity.

**\*For correspondence:**
marder@brandeis.edu

**Competing interest:** The authors declare that no competing interests exist.

## Editor's evaluation

Morozova et al., describe potential mechanisms contributing to the flexibility of burst patterns and dynamic responses to perturbations within an isolated reciprocally inhibitory circuit derived from the stomatogastric ganglion of the crab. The authors use the dynamic clamp approach to study the interactions between pharmacologically isolated, intrinsically silent gastric mill neurons. The authors demonstrate that the mechanisms of oscillation of the half-center networks are not fixed and shift to favor a release or escape mechanism depending on the synaptic threshold, IH conductance, and synaptic conductance. They also show that the different mechanisms of oscillation are differentially sensitive to neuromodulation and temperature changes. This is a fundamentally important study because reciprocally organized networks are ubiquitous and found virtually in every organism.

## Introduction

Reciprocal inhibition is ubiquitous in nervous systems, where it has many functions in sensory, motor, and cortical systems. Reciprocal inhibition between individual neurons or groups of neurons is the 'building block' of most half-center oscillators that generate antiphase and multiphase activity patterns (*Arbas and Calabrese, 1987a*; *Arbas and Calabrese, 1987b*; *Brown, 1997*; *Calabrese, 1998*; *Getting, 1989*; *Marder and Calabrese, 1996*; *Perkel and Mulloney, 1974*; *Sakurai and Katz,*

*2016*; *Satterlie, 1985*; *Soffe et al., 2001*; *Zang et al., 2020*). Due to their well-defined output, small reciprocally inhibitory circuits provide an excellent platform for investigating the resilience of circuits to internal and environmental challenges.

Theoretical studies have described two fundamentally different mechanisms of antiphase oscillations in half-center circuits: 'release' and 'escape' (*Skinner et al., 1994*; *Wang and Rinzel, 1992*). In the release mode the active cell falls below its synaptic threshold, thus, releasing the inhibited cell. In escape, the inhibited cell depolarizes above its synaptic threshold, thus, terminating the firing of the active cell. Whether the oscillator exhibits the escape or release mechanism depends on the position of the synaptic threshold within the slow-wave envelope of the membrane potential oscillation (*Skinner et al., 1994*; *Wang and Rinzel, 1992*). Many factors affect neuronal membrane potential and the synaptic threshold, including neuromodulators, temperature, and changes in the composition of the extracellular fluid. Perturbations can alter the balance of escape and release mechanisms and can create mixed mode oscillations. Although it is known that half-center oscillators can operate with a mixed mechanism (*Angstadt and Calabrese, 1989*; *Angstadt and Calabrese, 1991*; *Calabrese et al., 2016*; *Hill et al., 2001*), these have been less studied than oscillators in the pure release or pure escape mechanisms. Here, we also look at the increased or decreased resilience of oscillators operating in a mixed regime.

Some of the theoretical predictions of how oscillations are generated and controlled in reciprocally inhibitory circuits were tested in biological neurons in the crab stomatogastric ganglion (STG) by *Sharp et al., 1996* and *Grashow et al., 2009*, and in the leech heartbeat circuit (*Olypher et al., 2006*; *Sorensen et al., 2004*). These authors used the dynamic clamp, which utilizes a real-time computer interface to simulate nonlinear voltage-dependent synaptic and intrinsic currents in biological cells. *Sharp et al., 1996* studied the effects of varying computer-generated parameters on the circuit output and confirmed theoretical predictions that the switch in the mechanism of oscillations in a biological network is possible by shifting the synaptic threshold. *Grashow et al., 2009* extended their work by studying the effects of the neuromodulators, oxotremorine and serotonin, on the dynamic clamp created half-center networks. They observed a substantial variability in individual circuit responses to neuromodulation. *Sorensen et al., 2004* and *Olypher et al., 2006* studied the regulation of rhythmic bursting in a hybrid system of leech heartbeat interneurons by intrinsic currents, such as the H current and low-threshold $Ca^{2+}$ current.

Most theoretical studies on half-center oscillators were done with two identical neurons (*Daun et al., 2009*; *Nadim et al., 1995*; *Skinner et al., 1994*; *Wang and Rinzel, 1992*; *Zhang and Lewis, 2013*) with the notable exception of *Onasch and Gjorgjieva, 2020*. In some biological systems, half-center oscillators are formed between pairs of neurons that are ostensibly 'identical' or are copies of the same neuron type, such as, in the leech heartbeat system or sea slug escape swimming central-pattern generators (CPGs) (*Katz, 2016*; *Marder and Calabrese, 1996*; *Sakurai and Katz, 2016*). That said, even when biological half-center oscillators are formed from the reciprocal inhibition of two neurons of the same cell type, there is always some variability between the two neurons. Reciprocal inhibition between different classes of neurons can also be crucial for the operation of central pattern generating or other circuits, such as in the stomatogastric ganglion (*Bartos et al., 1999*; *Blitz and Nusbaum, 2011*; *Marder and Bucher, 2007*; *Marder and Calabrese, 1996*). In this case, there is no presumption that the intrinsic properties of the two neurons are identical. In this paper, we exploit the biological variability between the neurons we study to examine the robustness of the half-center oscillator on the extent of asymmetry between the two neurons used to form the half-center oscillator.

Robustness can be simply defined as a system's ability to maintain its characteristic functional properties despite perturbation. That said, in each particular system studied it can be challenging to articulate which features are central to robustness. In some instances, one might consider robustness in terms of whether the system is insensitive to perturbations that might result in qualitative state changes. In other contexts, one might be interested to determine how insensitive a particular feature is to a perturbation. In this manuscript, we sometimes examine the likelihood that a perturbation will result in an entirely new circuit state. At other times, we will be asking questions about the relationship of a perturbation for specific circuit features, such as frequency, duty cycle, or number of spikes per burst. But in all cases, we are trying to capture whether a perturbation is likely to alter the qualitative function of the circuit and/or change its state.

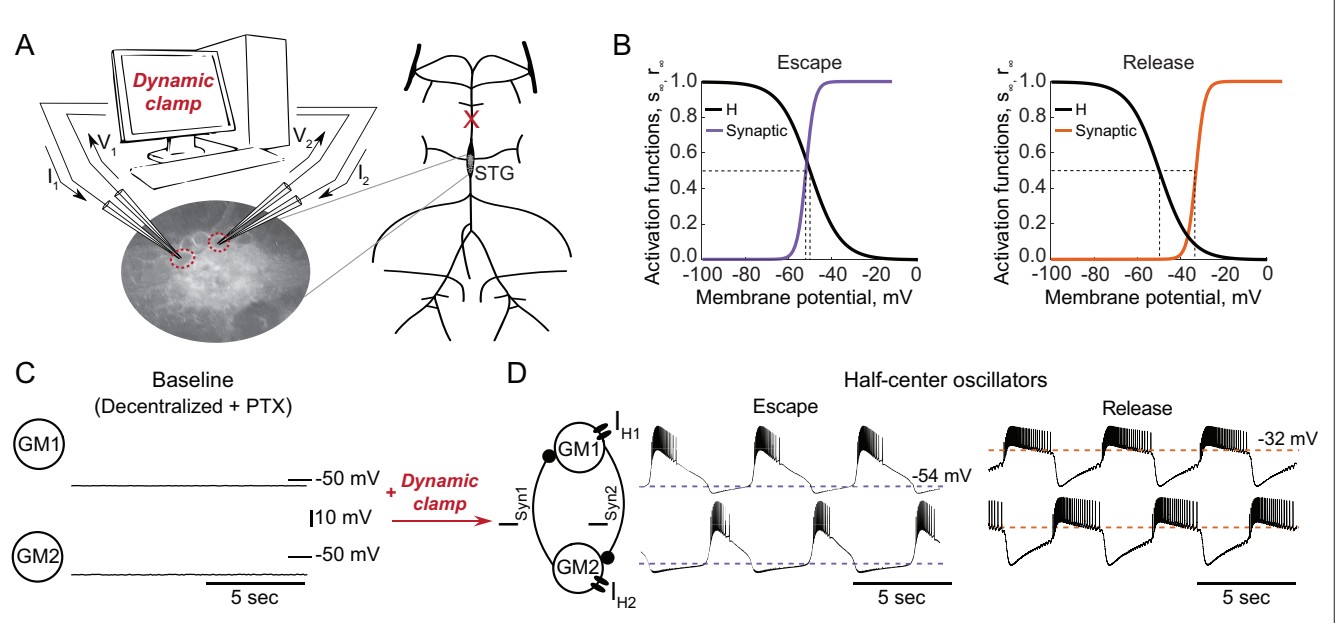

**Figure 1.** Experimental set-up. (**A**) Half-center oscillator circuits are built by connecting two gastric mill (GM) neurons from the stomatogastric ganglion (STG) of the crab *Cancer borealis* via artificial reciprocal inhibitory synapses ($I_{Syn}$) and by adding an artificial hyperpolarization-activated inward current ($I_H$) in two-electrode dynamic-clamp mode using RTXI. The membrane potentials of the neurons ($V_1$, $V_2$) are digitized and passed to a computer to calculate the currents ($I_1$, $I_2$), which are then converted to analogue signals and injected into the appropriate neurons. (**B**) Activation curves of the dynamic clamp generated H current and synaptic current. Shift in the synaptic activation curve switches the mechanism of oscillations between escape (left graph, purple curve) and release (right graph, orange curve). (**C**) At baseline, synaptically isolated GM neurons are silent with a resting membrane potential between –65 and –55 mV. (**D**) When coupled via the dynamic clamp, the neurons generate an alternating bursting pattern of activity (half-center oscillator). Representative half-center oscillator traces with escape mechanism are shown on the left and with release mechanism on the right. Synaptic thresholds are indicated by the horizontal dashed lines. In the circuit diagram, filled circles indicate inhibitory synapses.

Individual neurons and the circuits they form can show a high level of degeneracy in their intrinsic and synaptic properties (*Goaillard and Marder, 2021*; *Sharp et al., 1996*, *Calabrese, 2021*; *Prinz et al., 2004*). Previous studies demonstrated that neuronal networks with similar underlying parameters that generate similar behavior can respond differently to perturbations (*Alonso and Marder, 2020*; *Prinz et al., 2004*; *Tang et al., 2012*). We performed dynamic clamp experiments of half-center oscillators with similar underlying parameters but different oscillatory mechanisms using temperature and neuromodulation as perturbations to address some of the following questions: Are circuits with different underlying mechanisms of oscillation equally robust to intrinsic and environmental perturbations? What are the factors that play a key role in immediate circuit resilience against perturbations? What role does the dynamical mechanism of oscillation play in the circuit responses to neuromodulation? How does asymmetry between the units forming a half-center oscillator affect the output of the circuit?

## Results

### The output of reciprocally inhibitory neurons is shaped by their intrinsic and synaptic properties

To explore the interactions between intrinsic and synaptic parameters underlying variability in circuit behaviors and differential robustness to perturbations, we used the dynamic clamp to build half-center oscillator circuits using pharmacologically isolated gastric mill (GM) neurons of the stomatogastric ganglion (STG) of the crab *Cancer borealis* (*Figure 1A*, Materials and methods). Half-center circuits were formed by connecting two neurons via artificial reciprocal inhibitory synapses and by adding hyperpolarization-activated inward (H) currents, following the methods described in *Sharp et al., 1996*. Activation curves for the synaptic and H currents are shown in *Figure 1B*.

GM neurons are silent in the absence of modulatory and synaptic inputs (*Figure 1C*) and fire tonically when depolarized. When coupled together via reciprocal inhibitory connections and with addition of H current via dynamic clamp they can generate an antiphase bursting pattern of activity (*Figure 1D*). There are two fundamental mechanisms of antiphase bursting in these circuits – 'release' and 'escape' (*Wang and Rinzel, 1992*; *Skinner et al., 1994*). The mechanism of oscillation depends on the position of the synaptic threshold within the slow-wave envelope of the membrane potential oscillations. Thresholds that are close to the most hyperpolarized portion of the slow-wave generate an escape mechanism, while high synaptic thresholds that are close to the top of the slow-wave envelope lead to a release mechanism (*Figure 1D*). In escape mode, the transition between on and off states is graded, because at the hyperpolarized synaptic thresholds, synaptic activation is nearly saturated at depolarized voltages, and provides very little contribution to the synaptic current. In release mode, on-off transitions largely depend on spike-mediated transmission, because the synaptic threshold is at the very top of the slow-wave depolarization. Thus, in release, spike amplitude and frequency play a key

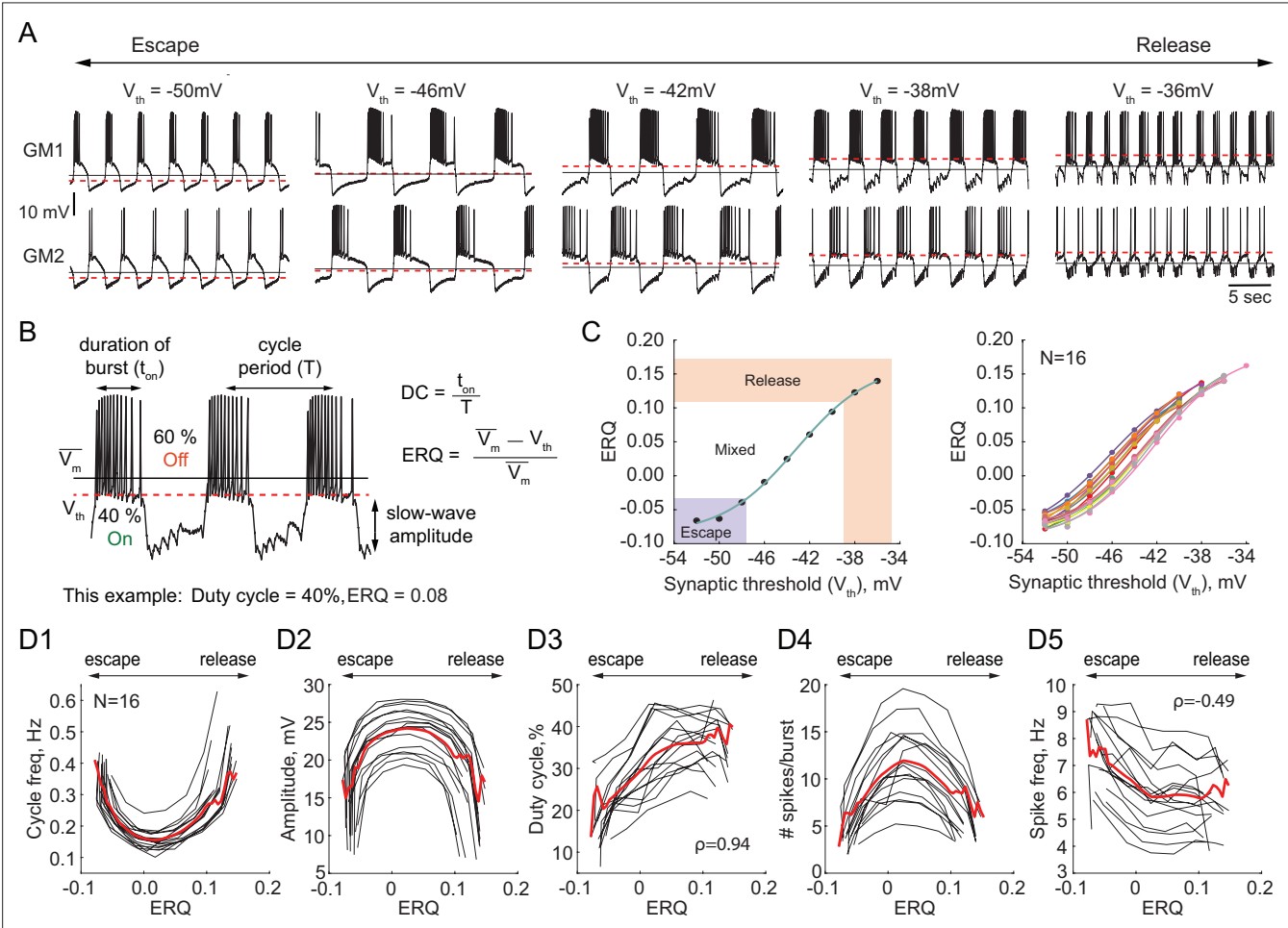

**Figure 2.** Dependence of the characteristics of half-center oscillator output on the mechanism of oscillation. (**A**) Representative intracellular recordings of GM neurons coupled via the dynamic clamp to form a half-center oscillator for different synaptic thresholds ($V_{th}$). Red dashed lines correspond to the synaptic thresholds, black solid lines correspond to the mean membrane potentials ($\overline{V_M}$). Depolarization of the synaptic threshold switches the mechanism of oscillations from escape to release passing through a mixture of mechanisms. (**B**) Half-center oscillator activity characteristics measured in this study, such as cycle period (frequency), slow-wave amplitude and duty cycle (DC) are indicated on the example GM neuron trace. Escape to Release Quotient (ERQ) is calculated based on the mean membrane potential and the synaptic threshold as shown. (**C**) ERQ as a function of the synaptic threshold for a single preparation (left) and multiple preparations (N = 16, right). Relationship between the ERQ and the synaptic threshold is sigmoidal as shown by the fit curve (cyan). Left hand ERQ plot is from the experiment shown in (**A**) (**D1**) Cycle frequency vs ERQ. (**D2**) Slow-wave amplitude vs ERQ. (**D3**) Duty cycle vs ERQ. (**D4**) Number of spikes per burst vs ERQ. (**D5**) Spike frequency vs ERQ. Black lines are individual experiments (N = 16), red lines represent means across all the experiments.

role in determining the properties of oscillation. By shifting the synaptic activation curve via dynamic clamp, we change the mechanism of oscillation between escape and release (**Figure 1B**).

## Characteristics of half-center oscillator output depend on the mechanism of oscillation

We investigated the dependence of the output of half-center oscillator circuits on the synaptic threshold while fixing the synaptic and H conductances ($g_{Syn}$ = 200 nS, $g_H$ = 300 nS). In each experiment we varied the synaptic threshold from –54 to –28 mV in 2 mV steps (N = 16, **Figure 2A**). We then characterized how physiologically relevant properties of the circuit output, for example, the cycle frequency, amplitude of oscillations, duty cycle, spike frequency and number of spikes per burst depend on the synaptic threshold (**Figure 2B**). Because of the inherent intrinsic differences in the biological neurons that comprise the half-centers, the same value of synaptic threshold does not necessarily generate the same mechanism of oscillation across preparations, as the relative position of the threshold within the slow-wave and the excitability of the neurons define the mechanism of oscillation. Thus, to quantitatively characterize the mechanism of oscillation across preparations and quantify the changes in the mechanism, we introduce a measure called Escape to Release Quotient (ERQ). This allowed us to characterize changes in the mechanism of oscillation in response to perturbations or changes in circuit parameters. We defined ERQ with the following equation:

$$ERQ = \frac{\bar{V_M} - V_{th}}{\bar{V_M}} .$$

$\bar{V_M}$ is a mean membrane potential averaged across both neurons in a circuit and $V_{th}$ is the synaptic threshold.

The left panel of **Figure 2C** shows that the relationship between the synaptic threshold and ERQ is well fit by a sigmoidal function ($R^2 = 0.998$). At the top of the sigmoid (above 0.11 in **Figure 2C**) the circuits are in a release mechanism. At the bottom of the sigmoid (below –0.033 in **Figure 2C**) the circuits are in an escape mechanism. The threshold ERQ values for release and escape were defined based on the maximum and minimum of the second derivative of the sigmoid functions that were fit to ERQ vs $V_{th}$ data for each experiment. The ERQ threshold for escape is -0.038 ± 0.008, while the ERQ threshold for release is 0.105 ± 0.012. The near-linear portion of the sigmoidal curve corresponds to a mixture of the mechanisms. The mixed regime demonstrates characteristics of both mechanisms with various balances between the mechanisms depending on the relative position of the threshold within the slow-wave envelope. The right panel in **Figure 2C** shows the dependence of the ERQ on the synaptic threshold across 16 preparations.

The cycle frequency shows a U-shaped relation as a function of the ERQ (**Figure 2D1**), as also seen in **Sharp et al., 1996**. The slow-wave amplitude shows an inverted U-shape dependence on the ERQ and is inversely correlated with the cycle frequency (**Figure 2D2**, Pearson correlation coefficient $r = -0.9$). The duty cycle (the burst duration divided by the cycle period) increases as the mechanism of oscillations changes from escape to release (**Figure 2D3**, $\rho = 0.94$, $p < 0.001$, Spearman rank correlation test). The difference in the duty cycle of circuits with different mechanisms can be explained by the difference in the magnitudes of the synaptic current. Because the synaptic threshold in escape is significantly more hyperpolarized relative to the release case, the magnitude of the synaptic current in a postsynaptic cell during its active phase is larger in the escape mechanism than in release, causing a steep hyperpolarization of the membrane potential below the neuron's spike threshold. The number of spikes per burst also shows an inverted U-shaped dependence on ERQ (**Figure 2D4**). The spike frequency decreases as the mechanism of oscillation changes from escape to release (**Figure 2**, $\rho = -0.49$, $p < 0.001$, Spearman rank correlation test). The higher spike frequency in escape mode is caused by a strong rebound current.

## Circuit output as a function of synaptic and H conductances in escape vs release

We investigated the dependence of the output of reciprocally inhibitory circuits on their synaptic ($g_{Syn}$) and H ($g_H$) conductances. In each experiment, we varied $g_{Syn}$ and $g_H$ from 150 nS to 1,050 nS in steps, mapping combinations of these parameters to characteristics of the output of the circuits operating

with escape or release mechanisms. The mechanism of oscillation for each map was determined based on the ERQ thresholds of -0.038 for escape and 0.105 for release that we established earlier. For the escape mechanism, the synaptic thresholds were between –54 and –50 mV across experiments. Variability in the synaptic thresholds for the escape mechanism comes from the variability in the resting membrane potentials of the neurons across preparations. For the release mechanism, synaptic thresholds were between –38 and –30 mV across experiments. Variability in the synaptic thresholds for the release mechanism comes from both the differences in the resting membrane potentials and the intrinsic excitability properties of the cells, such as spike thresholds, number of spikes per burst, and spike frequencies.

*Figure 3* summarizes pooled data from 20 experiments. Circuits operating in either release or escape produce stable alternating bursting which is distributed differently in the synaptic and H conductance space (*Figure 3A*). The gray scale in *Figure 3A* shows the fraction of bursting circuits operating with escape (left panel, N = 10) and release (right panel, N = 10) mechanisms at each $g_H$-$g_{Syn}$ parameter set. There are more circuits that generate half-center activity in release than in escape across these parameters. The synaptic and H currents must be tightly correlated to produce robust bursting in escape, but not in release mode. These findings suggest that half-center oscillators with a release mechanism are more robust to changes in either synaptic or H conductances, in terms of preserving their characteristic functional circuit state, compared to half-centers with an escape mechanism. In addition, these results provide a potential explanation of the across-preparation variability in conductance sets leading to stable bursting in reciprocally inhibitory circuits with a fixed synaptic threshold observed by *Grashow et al., 2009*.

*Figure 3B–F* characterizes the dependence of cycle frequency, oscillation amplitude, duty cycle, spike frequency and the number of spikes per burst on synaptic and H conductances. Increase in H current decreases the cycle frequency of the circuits in release (*Figure 3B*, right panel), but increases the cycle frequency in escape (*Figure 3B*, left panel). In escape, increasing the H conductance helps the inhibited neuron depolarize above the synaptic threshold faster, thus increasing the oscillation frequency. In release, increasing the H conductance prolongs the active phase of an uninhibited neuron, thus decreasing the frequency of oscillation. In both cases, the oscillation frequency decreases with the increase in inhibitory synaptic conductance (*Figure 3B*). The slow-wave amplitude (*Figure 3C*), number of spikes per burst (*Figure 3D*) and spike frequency (*Figure 3E*) decrease in the escape circuits but increase in the release circuits when H conductance is increased. The duty cycle is relatively independent of variations in synaptic and H conductances in either release or escape cases (*Figure 3F*, *Figure 3—figure supplement 1*). For all sets of $g_{Syn}$ and $g_H$, the duty cycles of the escape half-center oscillators are significantly lower than the duty cycles of the release half-center oscillators ($19.5 \pm 3.6\%$ in escape vs $42.4 \pm 3.3\%$ in release, * $p < 0.001$, Wilcoxon rank-sum test).

For a range of $g_H$-$g_{Syn}$ parameter sets, the characteristics of the output of half-center oscillators with escape and release mechanisms are not statistically different (*Figure 3B–E*, indicated by the black boxes, Wilcoxon rank-sum test, p > 0.05). Thus, similar circuit function can be produced by both escape and release mechanisms for the same values of synaptic and H conductances, although the duty cycles are more disparate than other measures of circuit performance. Importantly, if the mechanism is not known a priori, it is practically impossible to identify it only based on baseline spike output (e.g. in extracellular recordings) without perturbing the system.

## Circuits operating in a mixture of mechanisms

In some biological systems, half-center oscillators rely on a mixture of escape and release mechanisms to generate alternating bursting patterns of activity (*Angstadt and Calabrese, 1989*; *Angstadt and Calabrese, 1991*; *Calabrese et al., 2016*; *Hill et al., 2001*). Neuromodulators can shift the synaptic threshold, thus affecting the mechanism of oscillation in the circuit (*Li et al., 2018*). We explored how the oscillatory mechanism and characteristics of the circuit output mapped onto $g_{Syn}$-$g_H$ parameter space change as we changed the synaptic threshold. There can be a continuum of mechanistic interactions in half-center oscillator circuits, weighted by different mechanisms, with synaptic escape and release mechanisms at the extremes of this continuum. *Figure 4* shows the transformation of the $g_{Syn}$-$g_H$ maps of the network output and oscillatory mechanisms by moving the synaptic threshold from –50 mV to –30 mV in 5 mV steps. The mechanism of oscillation for each scenario was determined based on the ERQ thresholds of -0.038 for escape and 0.105 for release that we established earlier. The

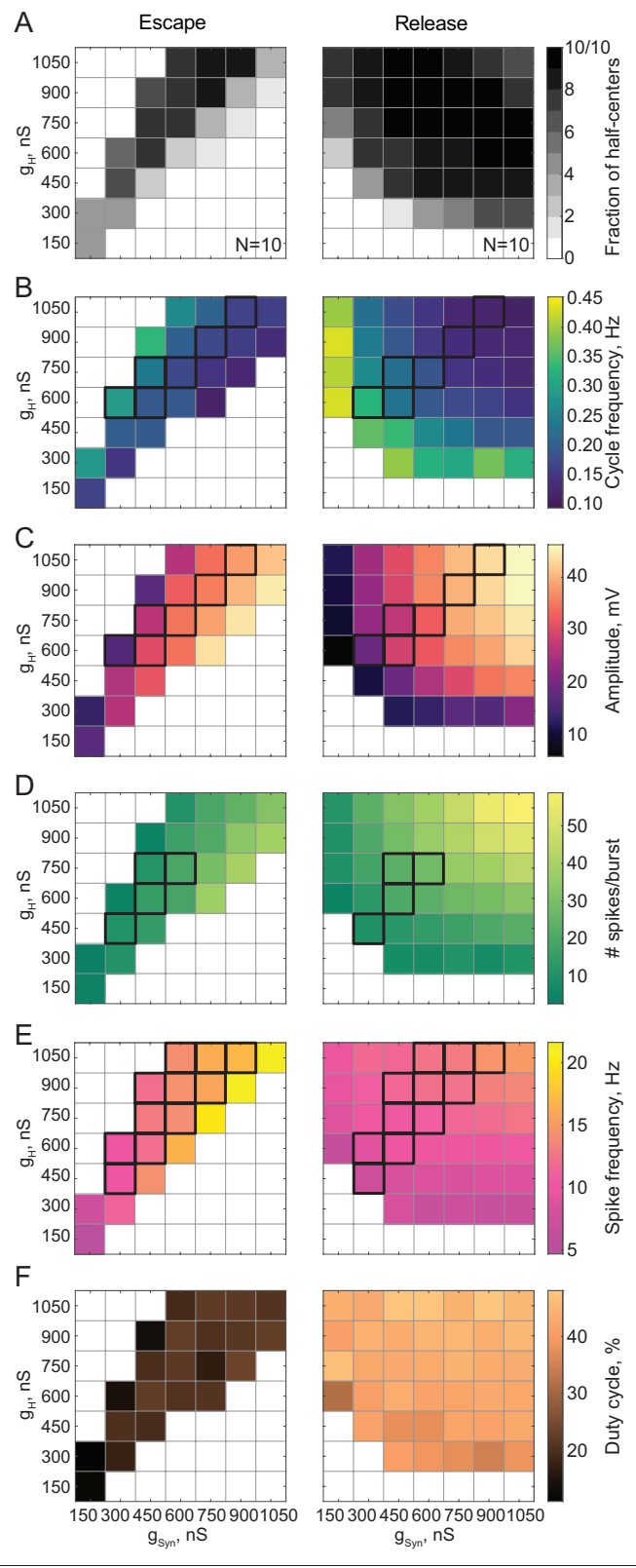

**Figure 3.** Maps of network output as a function of the synaptic and H conductances ($g_{Syn}$, $g_H$) for circuits with escape and release mechanisms. (**A**) Distribution of half-center oscillators in $g_{Syn}$-$g_H$ parameter space. Gray scale shows the fraction of preparations that formed half-center oscillators for each $g_{Syn}$-$g_H$ parameter combination within the map (N = 10 for each mechanism). White space corresponds to parameters sets for which no oscillators

*Figure 3 continued on next page*

*Figure 3 continued*

exist. (**B**) Dependence of the mean half-center oscillator cycle frequency on $g_{Syn}$ and $g_H$ across 10 preparations for each mechanism. (**C**) Dependence of the mean slow-wave amplitude on $g_{Syn}$ and $g_H$. (**D**) Dependence of the mean number of spikes per burst on $g_{Syn}$ and $g_H$. (**E**) Dependence of the mean spike frequency on $g_{Syn}$ and $g_H$. (**F**) Dependence of the mean duty cycle on $g_{Syn}$ and $g_H$. In panels B-E, $g_{Syn}$-$g_H$ parameter sets for which circuit output characteristics were not significantly different between release and escape are indicated by black boxes (Wilcoxon rank-sum test, $p > 0.05$). *Figure 3—figure supplement 1*. Dependence of the mean duty cycle of the circuits in escape, calculated based on time above synaptic threshold, on $g_{Syn}$ and $g_H$.

The online version of this article includes the following figure supplement(s) for figure 3:

**Figure supplement 1.** Dependence of the mean duty cycle of the circuits in escape, calculated based on time above synaptic threshold, on $g_{Syn}$ and $g_H$.

mechanism of oscillation is independent of $g_{Syn}$ and $g_H$ for the extreme cases of the hyperpolarized synaptic thresholds generating an escape mechanism (*Figure 4C and D*, left panel) and depolarized synaptic thresholds generating a release mechanism (*Figure 4C and D* right panel). Nonetheless, the mechanism is sensitive to the changes $g_{Syn}$ and $g_H$ for the intermediate values of the synaptic threshold, as evident by the substantial change in ERQ with $g_{Syn}$ and $g_H$ (*Figure 4A, C and D* middle panels).

*Figure 4A&B* depicts the ERQ and representative half-center voltage traces as a function of $g_{syn}$ and $g_H$ with a synaptic threshold of –40 mV. This network is in a mixture of escape and release for a wide range of $g_{syn}$ and $g_H$ The left-hand map illustrates a smooth transition in the balance of the mechanisms of oscillation as a function of changes in $g_{Syn}$ and $g_H$. The electrophysiological traces to the right illustrate the activity patterns at different map locations (*Figure 4B*). Increasing $g_H$ and decreasing $g_{Syn}$ biases the balance toward escape, while decreasing $g_H$ and increasing $g_{Syn}$ biases the mechanism towards release. Changing the mechanism of oscillation ultimately influences how the circuit will respond to stimuli and perturbations.

Theoretical studies have found that stable bursting is produced when the synaptic threshold is within the slow wave envelope of the membrane potential oscillations for circuits with graded synaptic transmission (*Skinner et al., 1994*). Thus, it might appear beneficial for the circuit to have a synaptic threshold in the middle, far from both the top and bottom of the slow wave. However, we observed that for the intermediate values of the synaptic thresholds ($V_{th}$=−45, –40, –35 mV, middle maps in *Figure 4C and D*), bursting is less regular (*Figure 4—figure supplement 1A*) and more asymmetric (*Figure 4—figure supplement 1B*). For the intermediate synaptic threshold of –40 mV, bursting exists for a small set of $g_{Syn}$-$g_H$ on the edge of the map, corresponding to weak synaptic coupling. This is because biological neurons, even of the same type, are never perfectly identical with respect to their intrinsic properties. Thus, the balance between the mechanisms is slightly different in the two cells, leading to situations when one of the cells does not have enough depolarizing drive to escape from inhibition, thus preventing the transition between the states. To illustrate the asymmetry in the activity, we calculated the ERQ values for the two neurons in a circuit independently and showed the associated mechanisms of oscillation. The ERQ values are slightly different between the neurons, indicating that neurons are making on-off transitions using different mechanisms for some combinations of synaptic and H conductances at intermediate values of the synaptic threshold (*Figure 4C and D*, colored outlines). For the extreme values of the synaptic thresholds both neurons operate with the same mechanism, either escape at –50 mV or release at –30 mV, despite the small differences in the ERQ values, resulting in more robust oscillations. Only a small subset of $g_{Syn}$-$g_H$ parameters allows for a smooth transition from one mechanism to another without losing alternating activity.

We characterized the dependence of cycle frequency, spike frequency, slow-wave amplitude, number of spikes per burst and duty cycle on synaptic and H conductances for different values of synaptic thresholds (*Figure 4E and F*, *Figure 4—figure supplement 1C*,D,E). For the synaptic threshold of –40 mV, the cycle frequency is independent of the change in H conductance (*Figure 4E* middle panel). The spike frequency increases with the increase in both $g_{Syn}$-$g_H$ for all the values of the synaptic thresholds (*Figure 4F*).

Besides alternating bursting pattern of activity, reciprocally inhibitory circuits can produce a rich array of other outputs, depending on the underlying parameters. We classified the activity patterns of reciprocally inhibitory circuits as either silent, asymmetric, irregular spiking, antiphase bursting or

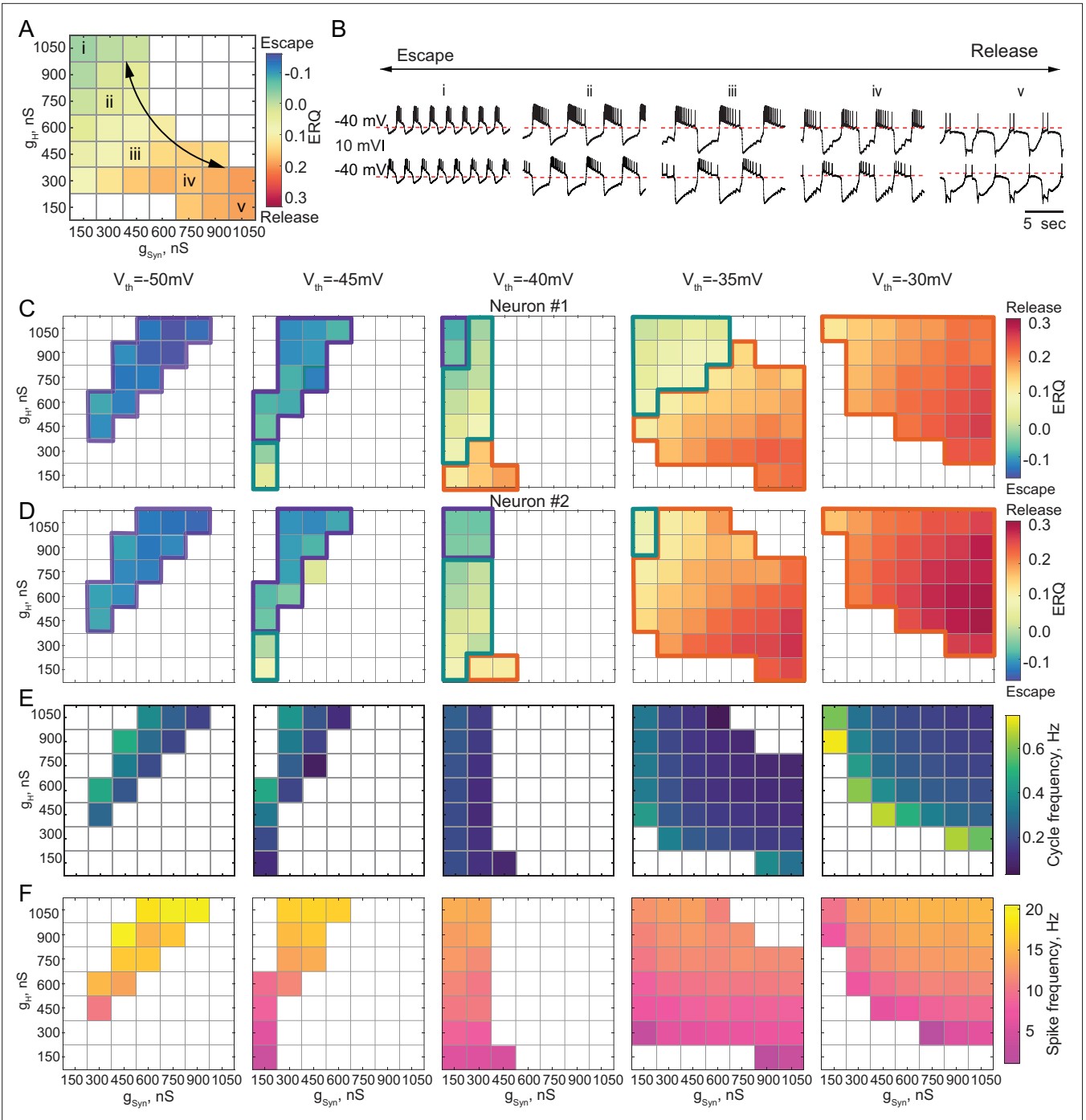

**Figure 4.** Dependence of the oscillation mechanism and half-center activity characteristics on the synaptic and H conductances for different synaptic thresholds. (**A**) ERQ as a function of synaptic and H conductances with a synaptic threshold of −40 mV in a single preparation. Mechanism of oscillation is sensitive to the changes in synaptic and H conductances at $V_{th}$=−40 mV: an increase in $g_{Syn}$ together with a decrease in $g_H$ switches the mechanism of oscillation from escape (top left corner in the map) to release (bottom right corner in the map). (**B**) Representative intracellular recordings of GM neurons coupled via the dynamic clamp corresponding to values of $g_{syn}$ and $g_H$ indicated in the parameter map (**A**) by roman numerals. (**C**) Dependence of ERQ on $g_{Syn}$ and $g_H$ for the synaptic thresholds of −50 mV, −45 mV, −40 mV, −35 mV, and −30 mV for one of the neurons in a circuit in a single preparation. Colored borders outline the regions of parameter space corresponding to different mechanisms of oscillation (escape-purple, release-orange or mixed-cyan). (**D**) Same as (**C**) but for the other neuron in a circuit. ERQ is relatively insensitive to changes in $g_{Syn}$ and $g_H$ in pure escape (left map) and pure release (right map) cases, but sensitive to $g_{Syn}$ and $g_H$ for intermediate thresholds (middle maps) similar to the experiment shown in panel (**A**). (**E**) Dependence of the half-center oscillator cycle frequency on $g_{Syn}$ and $g_H$ for different synaptic thresholds. (**F**) Dependence of the spike frequency on $g_{Syn}$ and $g_H$ for different synaptic thresholds.

*Figure 4 continued on next page*

*Figure 4 continued*

The online version of this article includes the following figure supplement(s) for figure 4:

**Figure supplement 1.** Output characteristics and activity patterns of reciprocally inhibitory circuits for different synaptic thresholds and different combinations of $g_{Syn}$ and $g_H$.

antiphase spiking for each set of $g_{Syn}$-$g_H$ and each value of the synaptic threshold (*Figure 4—figure supplement 1F*, see Materials and methods for the description of the classification algorithm). In the case of the escape mechanism, the circuits are typically silent or asymmetric for the parameter sets off the diagonal in the map (*Figure 4—figure supplement 1F* left panels). In contrast, in the case of release, the circuits typically show either antiphase or irregular spiking pattern of activity for low values of $g_{Syn}$ and $g_H$, on the border with antiphase bursting (*Figure 4—figure supplement 1F* right panel). For high values of $g_{Syn}$ and $g_H$, the circuit either shows antiphase bursting or asymmetric spiking (*Figure 4—figure supplement 1F* right panels), with one neuron constantly inhibiting the other one, depending on the asymmetry of neuronal intrinsic properties. The number of networks showing asymmetric firing pattern of activity is dominant on the $g_{Syn}$-$g_H$ map with the intermediate value of the synaptic threshold ($V_{th}$=-40 mV), uncovering the differences in the intrinsic properties of the half-center neurons (*Figure 4—figure supplement 1F* middle panel). This analysis allows us to predict how the activity pattern of reciprocally inhibitory circuits will change with the change of synaptic and H conductances, depending on the mechanism of oscillation.

## Effect of temperature on half-center oscillator circuits with temperature-independent synaptic and H currents

Rhythmic circuits, especially central pattern generators, must be robust to a wide range of global perturbations. Temperature is a natural and nontrivial perturbation that affects all biological processes to various degrees. We assessed the response of reciprocally inhibitory circuits relying on different mechanisms of oscillation to temperature changes. The dynamic clamp allowed us to study temperature-induced changes in the circuit output while isolating the effects of temperature on the synaptic and H currents from its effects on the cell-intrinsic currents. We built half-center oscillators with escape and release mechanisms and increased temperature in a smooth ramp from 10°C to 20°C (*Figure 5A*: release, 5B: escape). These temperatures were chosen based on the temperatures that *C. borealis* experiences in the wild. In the first sets of experiments, we intentionally kept the artificial synaptic and H currents temperature-independent to explore the role of temperature-induced changes in the intrinsic properties of the cells on the circuit output (case 1).

Reciprocally inhibitory circuits with a release mechanism become less robust as the temperature increases, as evident by a significant reduction in the slow-wave amplitude and increase in irregularity in the cycle frequency (*Figure 5A*). *Figure 5—figure supplement 1* illustrates an increase in the Coefficient of Variation of cycle frequency (*$p = 0.01$, Wilcoxon signed rank test) and spike frequency (*$p = 0.03$, Wilcoxon signed rank test) of circuits in release at higher temperatures. 9/15 release circuits lost oscillations when the temperature was increased by 10°C from 10°C to 20°C. The cycle frequency of these circuits significantly increases with an increase in temperature despite no changes in the properties of synaptic or H currents (*Figure 5A1,A5*). On the other hand, circuits with an escape mechanism are extremely robust to an increase in temperature (*Figure 5*). The cycle frequency of these circuits is remarkably stable during the changes in temperature (*Figure 5B4-5*, *Figure 5—figure supplement 1*).

## Effect of temperature on the intrinsic properties of GM neurons

To explain the observed changes in the circuit output on the basis of the changes in temperature, we characterized the intrinsic properties of the GM neurons in response to changes in temperature. We measured the mean resting membrane potential of GM neurons and their responses to current steps at temperatures between 10°C and 20°C. The membrane potential of GM neurons significantly hyperpolarized as temperature was increased from 10°C to 20°C (*Figure 5C*, n = 30, * $p < 0.001$, Wilcoxon signed rank test). This alters the relative position of the synaptic threshold within the envelope of membrane potential oscillation that defines the oscillation mechanism.

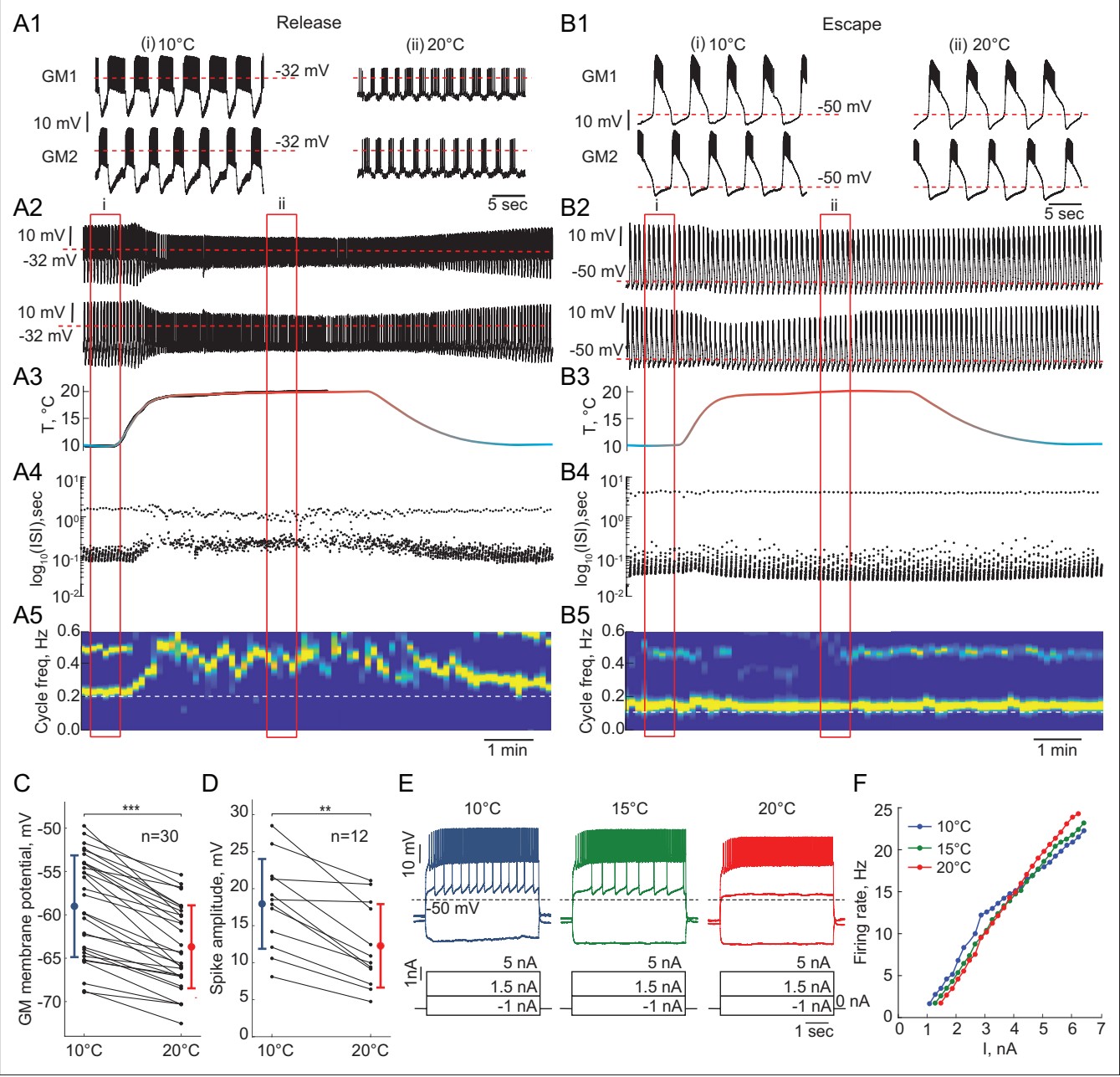

**Figure 5.** Response of reciprocally inhibitory circuits with release and escape mechanisms and temperature-independent artificial synaptic and H currents to an increase in temperature. (**A1**) 25 second segments of the activity of a half-center circuit with a release mechanism at 10°C and 20°C. (**A2**) Voltage traces of a half-center oscillator network in release during the increase in temperature for the entire representative experiment. (**A3**) Saline temperature. (**A4**) Inter spike intervals (ISI) of GM1 neuron during an increase in temperature plotted on a log scale. (**A5**) Spectrogram of the GM1 voltage trace, showing an increase in oscillation frequency at high temperature. Color code represents the power spectral density, with yellow representing the maximum power and blue the minimum power. Low-frequency band with the strongest power corresponds to the fundamental frequency of the periodic signal; secondary band at higher frequency corresponds to its 2f harmonic. (**B1-5**) Same as (**A1-5**) for a half-center oscillator circuit with an escape mechanism. (**C**) GM resting membrane potentials at 10°C and 20°C. for all the recorded neurons (n = 30). Each line corresponds to one neuron, colored circles and lines correspond to means ± standard deviation. Membrane potential of GM neurons is significantly more hyperpolarized at 20°C relative to 10°C (-59.0 ± 5.9 mV at 10°C vs -63.7 ± 4.8 mV at 20°C, *** p<0.0001, Wilcoxon signed rank test). (**D**) GM spike amplitudes at 10°C and 20°C measured at –40 mV in response to a current step for all the neurons (n = 12). The amplitude of GM spikes is significantly smaller at 20°C than at 10°C (17.9 ± 6.1 mV at 10°C vs 12.3 ± 5.6 mV at 20°C, *** p=0.0005, Wilcoxon signed rank test). (**E**) Representative voltage traces from a single GM neuron in response to current steps recorded at 10°C (blue), 15°C (green) and 20°C (red). (**F**) Frequency-current (f-I) relationships at 10°C (blue), 15°C (green) and 20°C (red) of the neuron from the representative experiment in panel (**E**).

*Figure 5 continued on next page*

*Figure 5 continued*

The online version of this article includes the following figure supplement(s) for figure 5:

**Figure supplement 1.** Coefficient of Variation (CV) of output characteristics of circuits in release and escape at low and high temperatures.

Spike amplitude, measured when the neurons were depolarized to –40 mV, decreased significantly with the increase in temperature from 10°C to 20°C (*Figure 5D*, n = 12, * $p < 0.001$, Wilcoxon signed rank test). This decrease in the spike amplitude decreases the robustness of half-center oscillators in a release mechanism, increasing the likelihood of a transition to a silent state, because at depolarized synaptic thresholds, the spikes provide a major contribution to the accumulation of synaptic current. In line with this, when spikes were blocked by TTX, the range of stable alternating activity was significantly reduced and dominated by synaptic escape (*Sharp et al., 1996*). Finally, we measured frequency-current (f-I) relationships (n = 13) and voltage-current relationships (V-I, n = 9) of GM neurons between 10°C and 20°C. *Figure 5E* shows representative voltage traces from a single GM neuron in response to current steps at 10°C, 15°C and 20°C. More current was needed to initiate spiking in GM neurons at higher temperatures (*Figure 5F*). The f-I curves became steeper at higher temperatures (*Figure 5F*, n = 13, 3.8 ± 1.6 Hz/nA at 10°C vs 4.8 ± 1.8 Hz/nA at 20°C, * $p = 0.013$, Wilcoxon signed rank test). There was no significant difference in the input resistance of GM neurons, measured by injecting negative current steps, at 10°C and at 20°C (n = 10, $p = 0.32$, Wilcoxon signed rank test). The changes in the intrinsic properties of GM neurons with temperature, that is hyperpolarization of membrane potential and decrease in the spike amplitude, are similar to previously reported changes in other neurons, including locust flight neurons (*Xu and Robertson, 1994*; *Xu and Robertson, 1996*), and *C. borealis* Lateral Gastric (LG) neurons (*Städele et al., 2015*).

Taken together, a combination of two factors: a relative depolarization of the synaptic threshold due to membrane potential hyperpolarization and a decrease in the spike amplitude, causes a loss of oscillations in the circuits with a release mechanism at high temperatures. At higher temperatures, the synaptic threshold becomes more depolarized than the top of the envelope of membrane potential oscillations, so that the transition between the active and inhibited states is governed by spiking activity (*Figure 5A*). In turn, a decrease in the spike amplitude leads to a decrease in the amplitude of the synaptic current, smaller hyperpolarization of a postsynaptic neuron, and, thus, smaller activation of H current in the postsynaptic neuron, decreasing the robustness of the oscillations. The difference in robustness of the circuits with a release mechanism is partially due to the individual variability in the sensitivity of the intrinsic properties of GM neurons to temperature changes.

While circuits with an escape mechanism that are comprised of neurons with similar intrinsic properties remain robust to an increase in temperature (*Figure 5B*), circuits comprised of the neurons with substantially different intrinsic excitability properties often 'crash' when the temperature increases. In the intrinsic escape mechanism, the ability of the neuron to depolarize above synaptic threshold and escape the inhibition relies on its intrinsic excitability. If one of the neurons is much less excitable than the other neuron it will be constantly suppressed by the more excitable neuron, not allowing the transition between the states to occur.

## The role of temperature-dependence of synapses and H current in the behavior and robustness of reciprocally inhibitory circuits

To study the effect of temperature-dependence in the parameters of the synaptic and H currents on the circuit responses to temperature, we implemented the temperature-dependence (1) only in synaptic and H conductances (case 2), (2) in both conductances and activation rates of the synaptic and H currents (case 3). *Figure 6* illustrates the behavior of representative escape and release circuits in response to gradual temperature increases in all the cases, including the case of temperature-insensitive synaptic and H currents for a comparison (right panels of *Figure 6*). The top panels of *Figure 7* show the percent change in cycle and spike frequencies of the representative circuits from *Figure 6*. The bottom panels of *Figure 7* show a summary of the effects of increasing temperature on multiple characteristics of circuit outputs across all experimental conditions (N = 33). The case of temperature-independent synapses and H current (case 1) is described in detail in the previous section and is summarized in *Figure 7* along with the other cases. All statistical tests, significance

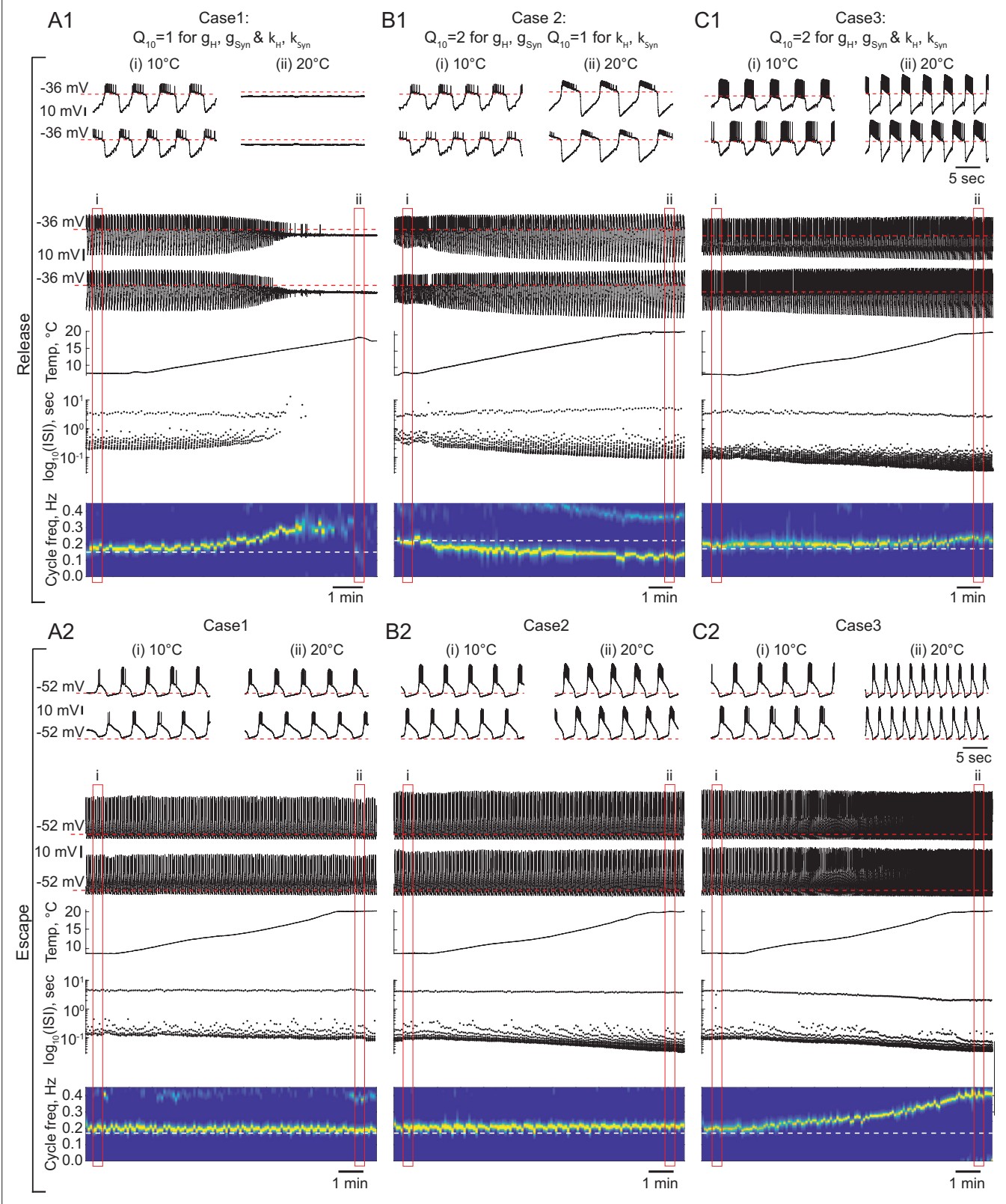

**Figure 6.** The role of temperature dependence in the synaptic and H currents in the response of the circuits with release and escape mechanisms to changes in temperature. (**A1**) Representative example of the behavior of a half-center oscillator in release in case of temperature-independent synaptic and H conductances and activation rates of these currents ($g_H$, $g_{Syn}$, $k_H$, $K_{Syn}$ $Q_{10} = 1$). Figure follows the same format as *Figure 5A–B*. (**A2**) Same condition as in (**A1**) for a circuit in escape. (**B1**) Representative example of the behavior of a half-center oscillator in release in case of temperature-

*Figure 6 continued on next page*

Figure 6 continued

dependence of the synaptic and H conductances with a $Q_{10} = 2$ and temperature-independent activation rates ($k_H$, $K_{Syn}$ $Q_{10} = 1$). (**B2**) Same condition as in (**B1**) for a circuit in escape. (**C1**) Representative example of the behavior of a half-center oscillator in release in case of temperature-dependence of the synaptic and H conductances and activation rates with a $Q_{10} = 2$. (**C2**) Same condition as in (**C1**) for a circuit in escape.

analyses, number of circuits/neurons and other relevant information for data comparison are provided in **Supplementary file 1a-1h**.

## Case 2: $Q_{10} = 2$ for the conductances and $Q_{10} = 1$ for the activation rates of the synaptic and H currents

We set the $Q_{10}$, a metric describing the ratio of rates of a biological process at two temperatures separated by 10°C, to two for the conductances of the synaptic and H currents (**Figure 6B1** release, 6B2 escape). $Q_{10} = 2$ is a typical value for experimentally measured $Q_{10}$s in STG neurons (**Tang et al., 2010**). Temperature driven increases in the conductances of the synaptic and H currents increase the amplitude of oscillations, thus, making the circuits with a release mechanism more robust in terms of maintaining its functional oscillatory state (**Figure 6B1** voltage traces, **Figure 7**). The cycle frequency of the circuits with a release mechanism decreases with an increase in temperature (**Figure 6B1**

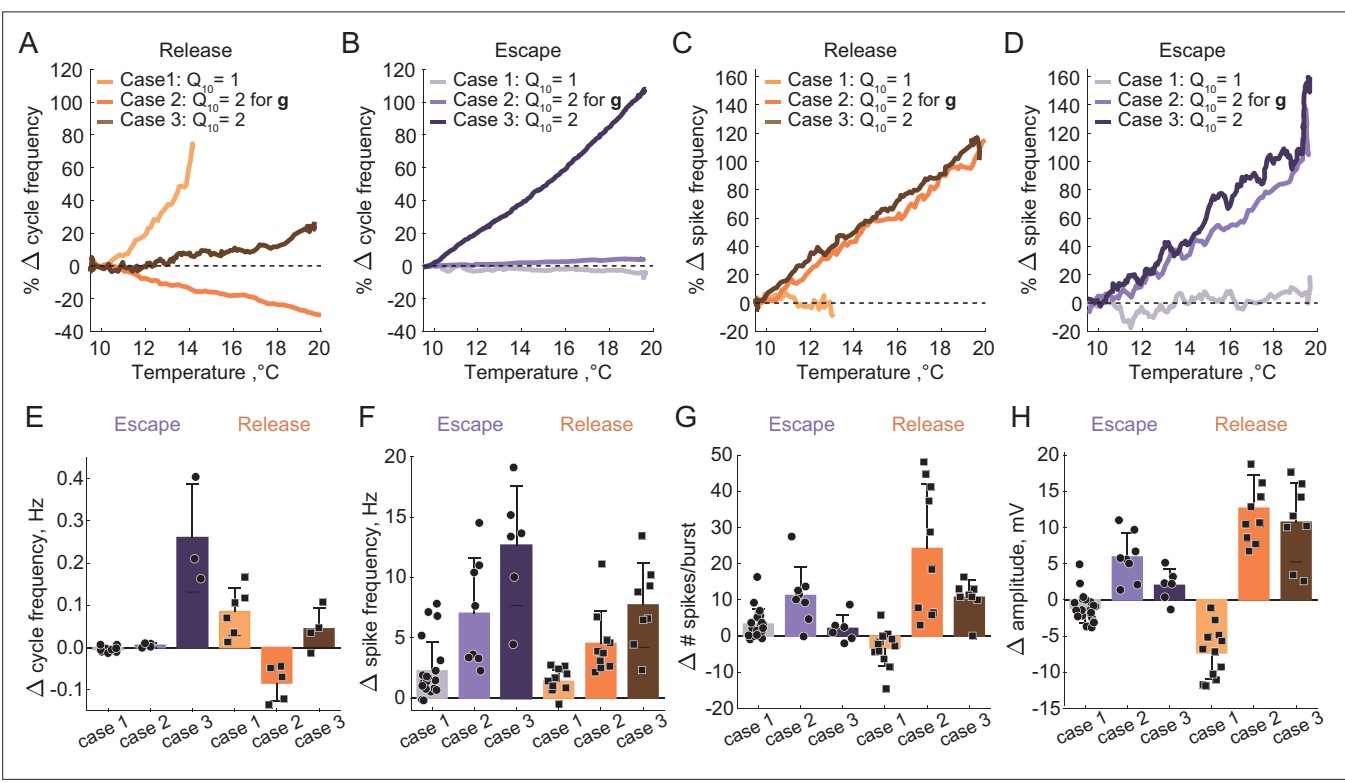

**Figure 7.** Summary of the effects of temperature on the characteristics of half-center oscillators with escape and release mechanisms and different temperature-dependences in the synaptic and H currents. (**A**) Percent change in cycle frequency of the release circuits shown in **Figure 6** with an increase in temperature from 10°C to 20°C. (**B**) Percent change in cycle frequency of the escape circuits shown in **Figure 6** with an increase in temperature from 10°C to 20°C. (**C**) Percent change in spike frequency of the release circuits in **Figure 6** with an increase in temperature from 10°C to 20°C. (**D**) Percent change in spike frequency of the escape circuits in **Figure 6** with an increase in temperature from 10°C to 20°C. (**E**) Change in cycle frequency with an increase in temperature from 10°C to 20°C across all experimental conditions (N = 33). (**F**) Change in spike frequency across all experimental conditions. (**G**) Change in number of spikes per burst across all experimental conditions. (**H**) Change in slow-wave amplitude across all experimental conditions. Case 1: $Q_{10} = 1$ for the conductances and the activation rates of the synaptic and H currents; Case 2: $Q_{10} = 2$ for the conductances and $Q_{10} = 1$ for the activation rates of the synaptic and H currents; Case 3: $Q_{10} = 2$ for the conductances and the activation rates of the synaptic and H currents.

The online version of this article includes the following figure supplement(s) for figure 7:

**Figure supplement 1.** Temperature can alter the mechanism of oscillations of reciprocally inhibitory circuits.

spectrogram, *Figure 7*), driven by the increases in both conductances in accordance with the findings shown in the right panel of *Figure 3B*. Temperature-dependence in the synaptic and H conductances makes circuits with a release mechanism more robust to an increase in temperature, by increasing the amplitude of oscillations.

The cycle frequency of the circuits with an escape mechanism remains constant over the whole temperature range (*Figure 6B2* spectrogram, *Figure 7B and E*), similar to the case of temperature-independent synapses and H current. Temperature-induced increases in the synaptic and H conductances counteract each other in the case of the escape mechanism as illustrated in the left panel of *Figure 3B*, (i.e. the frequency is conserved along the diagonal of $g_H$-$g_{syn}$ map). The spike frequency and number of spikes per burst of the circuits with either release or escape mechanisms significantly increase from 10°C to 20°C (*Figure 7C, D and H*).

## Case 3: $Q_{10} = 2$ for the conductances and the activation rates of the synaptic and H currents

We next implemented temperature-dependence in both the conductances and activation rates of the synaptic and H currents by setting these $Q_{10}s$ to 2. Both escape and release circuits maintained their oscillations as the temperature was increased in this case, due to the increase in the amplitude of the oscillations and faster transitions between the on-off states (*Figure 6C1* release, C2 escape). Although both circuits were bursting during the entire temperature range, there was a significant difference in the frequency responses of the escape and release circuits. Across all experiments, the cycle frequency of the circuit with a release mechanism did not significantly change over 10°C (*Figure 6C1* spectrogram, *Figure 7*, *Supplementary file 1a and b*), while the cycle frequency of the circuits with an escape mechanism increased dramatically (*Figure 6C2* spectrogram, *Figure 7*, *Supplementary file 1a and b*). In release, an increase in cycle frequency governed by changes in the intrinsic properties of the neurons and by an increase in the activation rates of synaptic and H currents was counteracted by a decrease in cycle frequency governed by an increase in synaptic and H conductances. The combination of these processes keeps the cycle frequency of release circuits nearly constant throughout the temperature ramp. In escape, an increase in cycle frequency is mostly driven by an increase of the activation rate of H current, because H current is causing the rebound. The spike frequency and the oscillation amplitude of circuit with either release or escape mechanisms significantly increased over 10°C, similar to the case of $Q_{10} = 2$ for conductances only (*Figure 7C, D and F*, *Supplementary file 1a and b*). The number of spikes per burst of the escape circuits did not significantly change with the increase in temperature, unlike in the release circuits (*Figure 7G*, *Supplementary file 1a and b*).

Characteristics of the circuit output are differently sensitive to temperature increase depending on the mechanism of oscillation and $Q_{10}s$ of the synaptic and ionic currents. The duty cycle was relatively independent of variations in temperature in all the cases (*Supplementary file 1a and b*). To assess whether temperature affects the mechanism of oscillation we calculated the ERQ values at 10°C and 20°C for different $Q_{10}$ cases (*Supplementary file 1a*). The ERQ did not significantly change for the release circuits (case 1: 0.13 ± 0.06 at 10°C, 0.14 ± 0.05 at 20°C, $p = 0.136$; case2: 0.16 ± 0.07 at 10°C, 0.15 ± 0.08 at 20°C, $p = 0.286$; case 3: 0.14 ± 0.07 at 10°C, 0.14 ± 0.08 at 20°C, $p = 0.575$, Wilcoxon signed rank test). The ERQ became significantly more positive for the escape circuits with temperature-independent synaptic and H currents, indicating the change in the mechanism of oscillation toward release with the increase in temperature (case 1: -0.09 ± 0.03 at 10°C, -0.07 ± 0.04 at 20°C, ** $p = 0.009$; case 2: -0.07 ± 0.03 at 10°C, -0.09 ± 0.03 at 20°C, * $p = 0.017$; case 3: -0.08 ± 0.05 at 10°C, -0.10 ± 0.04 at 20°C, $p = 0.173$, Wilcoxon signed rank test). An example of the change in the mechanism of oscillation from a mixture of intrinsic escape and synaptic release at 10°C all the way to a pure release mechanism at 20°C is shown in *Figure 7—figure supplement 1*. During the transition, the half-center exhibited characteristics of both mechanisms with various balances between the mechanistic interactions at different temperatures. The cycle frequency remained constant for a wide range of temperatures until the on-off transitions in the circuit were dominated by the synaptic release mechanism (*Figure 7—figure supplement 1F*).

## Effect of a neuromodulatory current on the robustness of circuits with release and escape mechanisms

A number of neurotransmitters and peptides converge on an inward current with the same voltage dependence, known as $I_{MI}$ (*Swensen and Marder, 2000*; *Swensen and Marder, 2001*). To explore the effect of $I_{MI}$ on reciprocally inhibitory circuits with different mechanisms of oscillation, we injected artificial $I_{MI}$ via the dynamic clamp into both neurons comprising half-center oscillators (*Figure 8A*). We then varied the synaptic threshold to alter the mechanism. *Figure 8B* illustrates representative recordings of a half-center oscillator at three different synaptic thresholds corresponding to escape, mixture, and release mechanisms in control (black traces) and with the addition of $I_{MI}$ (blue traces). We calculated the frequency of oscillations as a function of the synaptic threshold in control and with the addition of $I_{MI}$ ($g_{MI}$ = 150 nS). *Figure 8C* shows this relationship for the representative experiment in panel B. Artificially injected $I_{MI}$ produced no effect on the cycle frequency of escape circuits, while $I_{MI}$ decreased the cycle frequency of the circuits with a mixture of mechanisms or in release. Addition of $I_{MI}$ increased the robustness of circuits with a release mechanism, increasing the amplitude of oscillations (*Figure 8B*, right most traces) and expanding the range of synaptic thresholds producing stable antiphase bursting pattern of activity (*Figure 8C*). At the same time, $I_{MI}$ made oscillations less stable and irregular for circuits operating with a mixture of mechanisms, as evident by the increase in the CV of cycle frequency with addition of $I_{MI}$ (*Figure 8—figure supplement 1*, 0.06 ± 0.02 in control vs 0.13 ± 0.08 with $I_{MI}$, * p = 0.04, paired-sample Wilcoxon rank test) $I_{MI}$ amplified the asymmetry between the units comprising the circuit (*Figure 8B* middle traces, *Figure 8—figure supplement 1B*), and increased the standard deviation of the cycle frequency and elicited break in the central region of the cycle frequency curve corresponding to a mixed regime (*Figure 8C*). In the cases when the neurons had similar number of spikes per burst in control conditions, addition of $I_{MI}$ did not destabilize the circuits with the mixture of mechanisms. In the cases when the neurons had substantially different numbers of spikes in control, $I_{MI}$ amplified this difference (*Figure 8—figure supplement 1B*).

We quantified the change in cycle frequency, oscillation amplitude, duty cycle, spike frequency and number of spikes per burst across both neurons in circuits with the addition of modulatory current (N = 8, *Figure 8D1-D5*). All statistical tests and significance analyses of these data are provided in the legend of *Figure 8*. The cycle frequency of escape circuits did not change with the addition of $I_{MI}$ but significantly decreased in release circuits (*Figure 8D1*). $I_{MI}$ increased the amplitude of oscillations in both modes, with a significantly larger increase in release (*Figure 8D2*), making the oscillations more robust in terms of maintaining its functional oscillatory state. The duty cycle of the circuits in escape was statistically invariant to modulation, while there was a small but statistically significant increase in the duty cycle of the circuits in release (*Figure 8D3*). The number of spikes per burst significantly increased with $I_{MI}$ in release but not escape (*Figure 8D4*). Finally, $I_{MI}$ produced a small but statistically significant increase in the frequency of the spikes within bursts for both types of circuits (*Figure 8D5*). Overall, across all the characteristics, circuits with a release mechanism were significantly more sensitive to a modulatory current than circuits with an escape mechanism.

These observations suggest that the same type of modulation can produce different effects on the output of a circuit depending on the underlying mechanism of oscillation, and can make a circuit more or less susceptible to subsequent perturbations, potentially changing its sensitivity to pharmacological agents. For example, $I_{MI}$ increases the robustness of the circuit perturbed by an increase in temperature by preserving the oscillations (*Figure 8E*). In 4/4 preparations, $I_{MI}$ restored the antiphase oscillations in release circuits at high temperature, by depolarizing the neurons over the synaptic threshold and increasing the amplitude of oscillations. This is similar to the neuromodulatory rescue of the temperature-induced cessation of the gastric mill rhythm (*Städele et al., 2015*). This could be one of the mechanisms by which neuromodulators help maintaining a circuit's functional state at different temperatures.

## Discussion

One of the most difficult problems facing systems neuroscience is to determine the mechanisms that generate a given circuit output. The present work is designed to provide some fundamental insights into that problem, by studying a purposefully simple rhythmic circuit. Because some of the circuit parameters are constructed with the dynamic clamp, and are therefore known, we have been able to

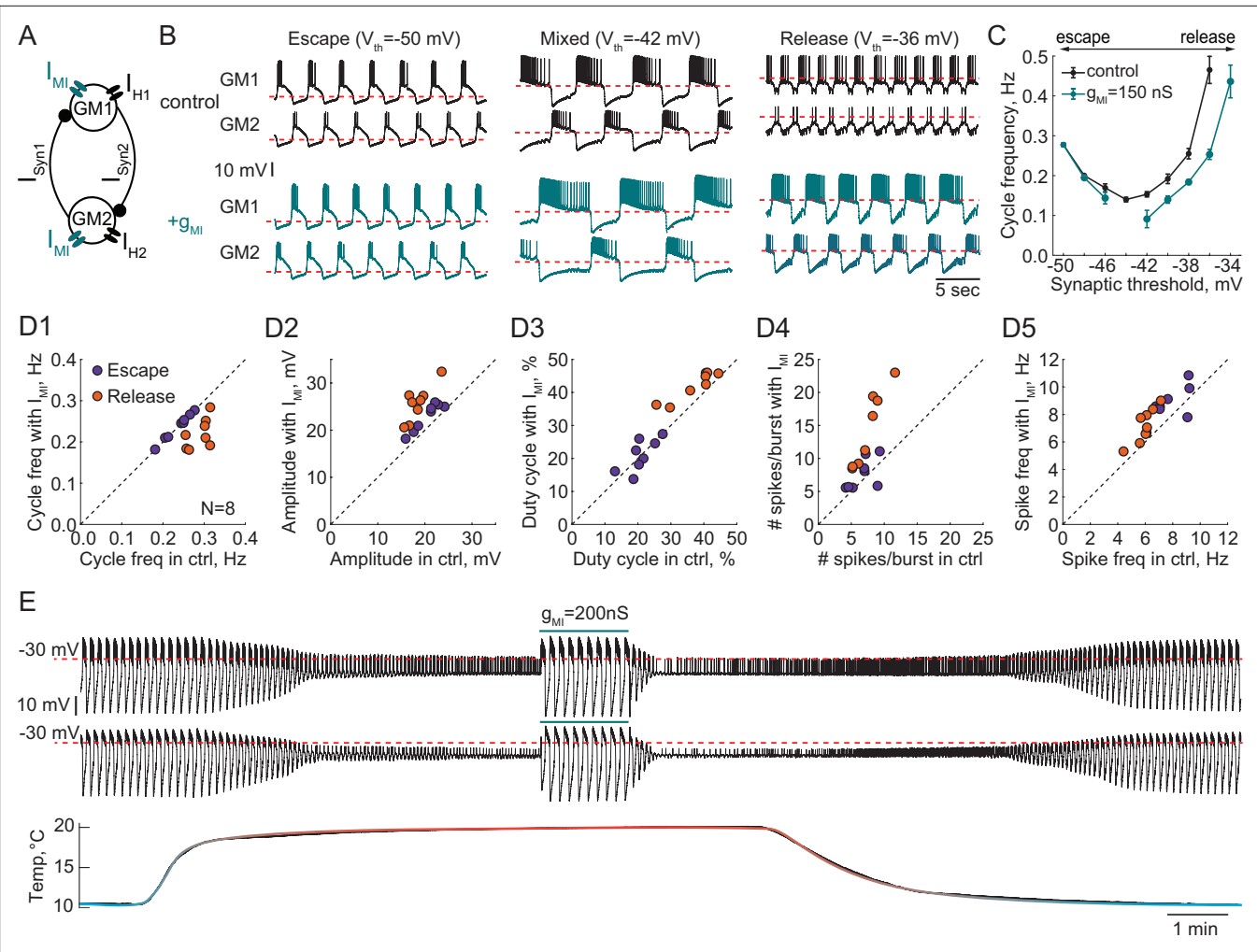

**Figure 8.** Effect of a modulatory current ($I_{MI}$) on the behavior of reciprocally inhibitory circuits with different oscillatory mechanisms. (**A**) A schematic representation of a reciprocally inhibitory circuit with a dynamic clamp modulatory current ($I_{MI}$). (**B**) Representative traces of a half-center oscillator for different synaptic thresholds in control (black) and with the addition of $I_{MI}$ ($g_{MI}$ = 150 nS, blue). (**C**) Oscillation frequency of the circuit in panel B as a function of the synaptic threshold in control (black) and with the addition of $I_{MI}$ (blue). (**D**) Characterizing the half-center oscillator output in escape and release with the addition of $I_{MI}$ (N = 8). (**D1**) Cycle frequency (Escape: 0.236 ± 0.033 Hz in control vs 0.237 ± 0.032 Hz with $I_{MI}$, n.s. p = 0.38, paired-sample t-test; Release: 0.289 ± 0.026 Hz in control vs 0.22 ± 0.036 Hz with $I_{MI}$, *** p = 0.0003, paired-sample t-test). (**D2**) Slow-wave amplitude (Escape: 20.5 ± 2.8 mV in control vs 23.0 ± 2.9 mV with $I_{MI}$, *** p < 0.0001, paired-sample t-test; Release: 18.4 ± 2.5 mV in control vs 25.7 ± 3.8 mV with $I_{MI}$, *** p < 0.0001, paired-sample t-test). Amplitude increase in release is significantly larger than in escape, *p = 0.02, paired-sample t-test. (**D3**) Duty cycle (Escape: 20.8 ± 4.3% in control vs 21.1 ± 4.9% with $I_{MI}$, n.s. p = 0.8, paired-sample t-test; Release: 37.3 ± 6.5% in control vs 42.2 ± 4.3% with $I_{MI}$, *p = 0.002, paired-sample t-test). (**D4**) Number of spikes per burst (Escape: 6.7 ± 1.9 in control vs 7.6 ± 2.3 with $I_{MI}$, n.s. p = 0.2, paired-sample t-test; Release: 7.6 ± 2.2 in control vs 14.4 ± 5.7 with $I_{MI}$, ** p = 0.001, paired-sample t-test). (**D5**) Spike frequency (Escape: 7.8 ± 1.2 Hz in control vs 8.7 ± 1.3 Hz with $I_{MI}$, * p = 0.029, paired-sample t-test; Release: 6.0 ± 0.8 Hz in control vs 7.2 ± 1.3 Hz with $I_{MI}$, ** p = 0.001, paired-sample t-test). (**E**) $I_{MI}$ restores the oscillations in the circuit with a release mechanism that stopped oscillating at high temperature. Example voltage traces of a half-center oscillator circuit in release during an increase in temperature from 10°C to 20°C. In this example, synaptic and H conductances and activation rates are temperature-independent.

The online version of this article includes the following figure supplement(s) for figure 8:

**Figure supplement 1.** Irregularity and asymmetry of oscillations.

gain insight into how circuits that appear similar in function can respond differently to the same perturbations. In dynamic clamp hybrid circuits, we have access to some of the hidden variables that define the dynamical mechanisms governing circuit behavior. At the same time, we have not sacrificed the complexity of the biological neurons. This allowed us to study how the interaction between biophysical and dynamical properties of these neural circuits define their robustness, by capturing how likely a perturbation is to alter the circuit's qualitative function and/or change its state. The findings of this

paper have implications for understanding animal-to-animal variability in circuit responses to various stressors and modulators.

There is a continuum of mechanistic interactions that produce the oscillatory transitions in the circuit, with pure synaptic escape and release on the ends of this continuum. A mixture of mechanisms is likely common in biological circuits, for example, in systems relying on both graded and spike-mediated transmission, such as in the leech heartbeat oscillator (*Angstadt and Calabrese, 1989*; *Angstadt and Calabrese, 1991*; *Calabrese et al., 2016*). However, the mechanism of oscillation in a given circuit is not a constant but can be altered by perturbations, modulators and/or changes in the circuit parameters. To quantitatively characterize the mechanism of oscillation and the degree to which it is affected by perturbations, we introduced a measure called the Escape to Release Quotient (ERQ). The ERQ quantifies the position of the synaptic threshold relative to the mean membrane potential of each neuron in the circuit. This measure is useful in defining the mechanism of oscillation because the position of the synaptic threshold within the membrane potential envelope, as well as the intrinsic properties of the neurons define the mechanism of oscillation (*Skinner et al., 1994*; *Sharp et al., 1996*). However, identifying the exact mechanism of oscillatory transitions is a difficult problem, because it requires knowing all of the underlying currents. Because some of the currents are computer-generated, we can describe their contributions to the transitions. In the mixed regime, on-off transitions are associated with a buildup of the H current in the inhibited cell, and a decay in the synaptic current. Decay in the synaptic current is associated with hyperpolarization of the membrane potential of the active cell. The degree to which changes in the synaptic and H currents contribute to the transitions depends on the position of the synaptic threshold within the slow wave.

The problem of identifying the mechanism of oscillation becomes more complicated if the neurons comprising the circuits are asymmetrical with respect to their intrinsic properties. In this case, the oscillatory transitions in each neuron can occur with different balances of mechanistic operations, and the mean ERQ value calculated across both neurons in the circuit does not accurately describe the mechanism of oscillation. This means that if the two neurons are appreciably different from each other, the ERQ must be calculated independently for each neuron to uncover the differences in their mechanisms. Finally, due to inherent intrinsic variability of the biological neurons that comprise the half-centers, the same parameters, i.e. the same value of the synaptic threshold, do not necessarily generate the same mechanism of oscillation across circuits. Thus, the ERQ, while an instructive measure is not a perfect way to characterize the complex mixtures of mechanisms, especially when the two biological neurons are different.

Unperturbed half-center circuits with escape and release mechanisms can have very similar characteristics, including burst and spike frequencies. Thus, if the mechanism is not known a priori, it is challenging to identify the underlying mechanisms of circuit function from the baseline spiking activity. One way to reveal hidden differences in the mechanism underlying circuit dynamics is by perturbing them. We showed that reciprocally inhibitory circuits with different underlying oscillation mechanisms are not equally robust to perturbations. Particularly, circuits in release mode are robust to variations in synaptic and H conductances, in terms of preserving their qualitative state, but sensitive to an increase in temperature and modulation. In contrast, the circuits in escape rely on tight correlations between synaptic and H conductances to generate bursting but are resilient to increases in temperature and modulation.

Previous computational studies showed that half-center oscillators relying on either release or escape mechanisms differentially respond to synaptic inputs and current pulses (*Hill et al., 2001*; *Nadim et al., 1995*; *Olsen et al., 1995*; *Zhang and Lewis, 2013*). *Daun et al., 2009* used model neurons with or without persistent sodium current to form half-center oscillators. When asymmetric noise was injected into only one of the neurons, half-centers operating in escape had a larger range of oscillation period than did circuits with release or a mixture of mechanisms (*Daun et al., 2009*). Additionally, half-centers built with two Morris-Lecar model neurons have significantly different phase response properties and phase locking dynamics depending on whether they operate in escape or release (*Zhang and Lewis, 2013*).

Oscillations in the leech heartbeat model rely on both graded and spike-mediated synaptic transmission and contain elements of both escape and release modes (*Hill et al., 2001*; *Nadim et al., 1995*; *Olsen et al., 1995*). The release mechanism in this model is caused by inactivation of slow calcium current, while the escape mechanism is promoted by activation of H current. Increase in H

conductance decreases the duration of the inhibited phase and increases the frequency of oscillations (*Olsen et al., 1995*). There is no single parameter in the leech heartbeat model that corresponds to a synaptic threshold, such as is implemented in this study. However, the maximal conductance of slow calcium current to some extent is comparable to the synaptic threshold, where higher conductances correspond to more hyperpolarized synaptic thresholds (*Olsen et al., 1995*). The interaction between the low-threshold ionic currents and the synaptic currents determines whether the escape or release mechanism is prevalent. Model half-center circuits are typically built with two identical neurons, although experimental data suggest that the conductance values and intrinsic properties of neurons even of the same type can differ significantly (*Doloc-Mihu and Calabrese, 2014*; *Goldman et al., 2001*; *Marder and Goaillard, 2006*; *Prinz et al., 2003*; *Prinz et al., 2004*; *Roffman et al., 2012*; *Schulz et al., 2006*; *Schulz et al., 2007*; *Srikanth and Narayanan, 2015*; *Swensen and Bean, 2005*; *Temporal et al., 2012*; *Tobin et al., 2009*; *Tran et al., 2019*). The studies in which the dynamic clamp is used to create half-center circuits from biological neurons profit from natural cell-to-cell and animal-to-animal variability to investigate circuit responses to stressors and modulators. *Sorensen et al., 2004* and *Olypher et al., 2006* built asymmetries into the circuit by unilateral variations of the parameters and found that despite the asymmetries and animal-to-animal variability in the underlying conductances, hybrid half-center oscillators built out of leech heartbeat interneurons were remarkably balanced. Interestingly, increasing asymmetries between the units did not destabilize the half-center oscillators. One possible explanation for the stability of the leech half-centers despite their asymmetries is that the individual neurons making up the half-center show distinct oscillatory properties that could aid in the stabilization. In contrast, the GM neurons from the crab STG are not intrinsically oscillatory but are silent or tonically spiking in the absence of their synaptic inputs. This may therefore make these circuits more sensitive to the asymmetries in the neurons forming the half-center.

*Grashow et al., 2009* found that either serotonin or oxotremorine (a muscarinic receptor agonist) increased the oscillation frequency of half-center oscillator circuits built with GM neurons and made alternating bursting more robust by extending the parameter range over which bursting exists. However, there was a substantial variability in individual responses of half-center circuits to neuromodulation, with a few circuits showing 'anomalous' decreases in cycle frequency in the presence of modulators. We speculate that some of the variability in *Grashow et al., 2009* may have been due to the differences in underlying mechanisms of oscillations across the circuits and the degree of asymmetry between the units comprising the circuit. We show that the same neuromodulatory current can either have no effect on a same circuit if operating in escape, destabilize the circuits if operating in mixed mode or expand the range of parameters producing stable bursting in a circuit if operating in release (*Figure 8*, *Figure 8—figure supplement 1*). Thus, knowing the dynamical mechanism involved in generating the circuit output is crucial for understanding the circuit responses to stimuli.

A similar variability in the response to a neuromodulator is seen in the crustacean gastric mill rhythm. This rhythm is generated by a half-center oscillator and can be elicited by multiple mechanisms (*Powell et al., 2021b*). Stimulation of the MCN1 projection neuron or bath-applying the peptide CabPK result in gastric mill rhythms with similar output patterns (*Powell et al., 2021b*). Despite the similarity of their baseline activity patterns, these rhythms rely on participation of different neurons and respond differently to hormone CCAP, which is known to activate $I_{MI}$ (*Swensen and Marder, 2000*; *Swensen and Marder, 2001*). CCAP slows down MCN1-generated rhythm but, in contrast, speeds up CabPK-generated rhythm (*Kirby and Nusbaum, 2007*; *Powell et al., 2021b*). We propose that the MCN1 rhythm might operate in release, while CabPK-rhythm operates in escape. Thus, different modulators can elicit different dynamical mechanisms of rhythm generation. In support of this hypothesis, it has been reported that similar gastric mill rhythms, which are generated by a stimulation of disparate neuromodulatory pathways, have different temperature sensitivity (*Powell et al., 2021a*; *Städele et al., 2015*). A modest temperature increase of 3°C abolishes the MCN1-rhythm (*Städele et al., 2015*), in contrast, the VCN-rhythm is temperature-robust over a wide range of temperatures, between 7°C and 25°C (*Powell et al., 2021a*). We propose that the difference in temperature sensitivity between the two versions of the gastric mill rhythm could be explained by the differences in their dynamical mechanisms of oscillation.

Many studies found significant correlations between the conductances of voltage-dependent currents in both invertebrates and vertebrates (*Amendola et al., 2012*; *Calabrese et al., 2011*; *Goaillard et al., 2009*; *Khorkova and Golowasch, 2007*; *Schulz et al., 2006*; *Schulz et al., 2007*). It has been argued that reliable circuit output and resilience to perturbations are enhanced by the

conductance correlations, rather than by the particular values of individual parameters (*Olypher and Calabrese, 2007*; *Onasch and Gjorgjieva, 2020*; *Tobin et al., 2009*; *Zhao and Golowasch, 2012*). In line with this, we found that synaptic and H conductances are positively correlated in the circuits with escape mechanisms (*Figure 3A*), contributing to the robustness of these circuits to variations in temperature. Changes in the synaptic and H conductances with temperature counteract each other keeping the oscillation frequency of escape circuits with temperature-independent activation-rates constant for a wide range of temperatures (*Figure 6B2*).

Because temperature differentially affects many nonlinear processes shaping circuit output, it is nontrivial for a circuit to maintain its function over a wide range of temperatures. Despite that, many neuronal circuits, including the pyloric and half-center driven gastric mill circuits of crustaceans, are temperature compensated and function over an extended physiological temperature range (*Haddad and Marder, 2018*; *Kushinsky et al., 2019*; *Powell et al., 2021a*; *Soofi et al., 2014*; *Tang et al., 2010*; *Tang et al., 2012*). Complicating the situation, circuit susceptibility to temperature changes is strongly influenced by the modulatory environment (*Haddad and Marder, 2018*; *Soofi and Prinz, 2015*; *Städele et al., 2015*). Obtaining insights into the mechanisms that underlie acute temperature resilience is difficult. Temperature is a particularly difficult perturbation to model in biologically plausible circuits because there are many free parameters to set, as temperature affects both the conductances and activation rates of the currents, making it a highly unconstrained problem. Because it is difficult to measure the temperature dependence of all of the currents in a given cell type (*Tang et al., 2010*), most modeling studies (*Alonso and Marder, 2020*; *Caplan et al., 2014*; *O'Leary and Marder, 2016*; *Rinberg et al., 2013*) employ $Q_{10}$ values that are only partially based on measured values. In simplified models it is possible to study the dynamical mechanisms of robustness and characterize bifurcations as a function of temperature (*Rinberg et al., 2013*), but many biophysical details are lost. In contrast, in the hybrid neural-computer dynamic clamp circuits studied in this paper, we can control the dynamical mechanisms governing circuit behavior and temperature-dependence in the computer-generated parameters, without making any assumptions about the temperature dependence of the intrinsic currents of the neurons. Thus, we benefit from not having to over-simplify the effects of temperature on the biological neurons.

It is as of yet unclear whether circuits that depend on one or another dynamical mechanism for operation are intrinsically more resilient to all perturbations, or whether robustness is determined idiosyncratically for each circuit configuration and perturbation. The present study illustrates how nontrivial it is to explain circuit function on the basis of basal firing pattern alone. The dynamical mechanisms underlying half-center oscillator transitions are well defined in modeling studies that reveal the underlying interactions between hidden state variables and voltage-dependent synaptic and intrinsic currents. While theoretical studies provide mechanistic insight, it can be quite difficult to establish how those mechanisms are instantiated in biological neurons. Moreover, virtually all previous computational studies in half-centers were done with identical neurons, and in no case will two or more biological neurons even of the same cell type, be identical. The dynamic clamp studies here provide access to some of the fundamental dynamical mechanisms important for generation of antiphase oscillations, while retaining the intrinsic 'features' of the biological neurons. In conventional current clamp experiments the investigator does not have a continuous access to state variables of the currents, while in the dynamical clamp experiments state variables of the computer-generated currents are readily accessible. A fundamental conclusion of this work is that very nuanced changes in circuit mechanism can profoundly alter circuit stability in response to perturbations and inputs. Thus, a challenge for the future will be developing new methods to extract dynamical mechanisms underlying circuit function from biological circuits while they are in operation.

## Materials and methods

### Key resources table

| Reagent type (species) or resource | Designation | Source or reference | Identifiers | Additional information |
|---|---|---|---|---|
| Biological sample (Jonah Crabs) | *Cancer borealis* (Jonah Crabs) Adult Male | Commercial Lobster (Boston, MA) | NCBI:txid39395 | |
| Chemical compound, drug | Picrotoxin (PTX) | Sigma-Aldrich | | |

*Continued on next page*

*Continued*

| Reagent type (species) or resource | Designation | Source or reference | Identifiers | Additional information |
|---|---|---|---|---|
| Chemical compound, drug | Tetrodotoxin (TTX) | Alamone labs | T-550 | |
| Software, algorithm | Dynamic clamp | Real-Time eXperiment Interface (RTXI) software versions 1.4 and 2.2 | http://rtxi.org/ | |
| Software, algorithm | pClamp version 10.5 | Molecular Devices, San Jose | https://www.moleculardevices.com/products/axon-patch-clamp-system/acquisition-and-analysis-software/pclamp-software-suite RRID: SCR_011323 | |
| Software, algorithm | MATLAB R2020a | MathWorks | https://www.mathworks.com/products/matlab.html | |
| Software, algorithm | IBM SPSS Statistics 24 | IBM | RRID:SCR_002865 | |
| Software, algorithm | Adobe Illustrator 2020 | Adobe | https://www.adobe.com/products/illustrator.html | |

## Animals and experimental methods

Adult male Jonah Crabs, *Cancer borealis*, (N = 43) were obtained from Commercial Lobster (Boston, MA) and maintained in artificial seawater at 10°C–12°C in a 12 hr light/dark cycle. On average, animals were acclimated at this temperature for 1 week before use. Prior to dissection, animals were placed on ice for at least 30 min. Dissections were performed as previously described (*Gutierrez and Grashow, 2009*). In short, the stomach was dissected from the animal and the intact stomatogastric nervous system (STNS) was removed from the stomach including the commissural ganglia, esophageal ganglion and stomatogastric ganglion (STG) with connecting motor nerves. The STNS was pinned in a Sylgard-coated (Dow Corning) dish and continuously superfused with saline. Saline was composed of 440 mM NaCl, 11 mM KCl, 26 mM $MgCl_2$, 13 mM $CaCl_2$, 11 mM Trizma base, 5 mM maleic acid, pH 7.4–7.5 at 23°C (~7.7–7.8 pH at 11°C).

## Electrophysiology

Intracellular recordings from the somata of gastric mill (GM) neurons were made using two-electrode current clamp in the desheathed STG with 10–20 MΩ sharp glass microelectrodes filled with 0.6 M $K_2SO4$ and 20 mM KCl solution (*Figure 1A*). Intracellular signals were amplified with an Axoclamp 900 A amplifier (Molecular Devices, San Jose). Extracellular nerve recordings were made by building wells around nerves using a mixture of Vaseline and mineral oil and placing stainless-steel pin electrodes within the wells to monitor spiking activity. Extracellular nerve recordings were amplified using model 3500 extracellular amplifiers (A-M Systems). Data were acquired using a Digidata 1,440 digitizer (Molecular Devices, San Jose) and pClamp data acquisition software (Molecular Devices, San Jose, version 10.5) and Real-Time eXperiment Interface (RTXI) software (http://rtxi.org/) version 2.2 or 1.4. Recordings were done with a sampling frequency of 10 kHz. For identification of GM neurons, somatic intracellular recordings were matched to action potentials on the dorsal gastric nerve (*dgn*), and/or the anterior lateral nerve (*aln*).

For the process of blocking descending modulatory inputs to the STG, a Vaseline well was built around the exposed portion of the *stn*. Propagation of axonal signaling, and, thus, neuromodulatory release, was blocked from upstream ganglia by replacing saline in the Vaseline well with $10^{-7}$M tetrodotoxin (TTX) in a 750 mM sucrose solution. $10^{-5}$M Picrotoxin (PTX) was used to block inhibitory glutamatergic synapses (*Marder and Eisen, 1984*). Preparations were allowed to stabilize after decentralization and PTX application for at least 1 hr prior to building a reciprocally inhibitory circuit via dynamic clamp.

## Dynamic clamp

To create the half-center oscillator circuits, artificial reciprocal inhibitory synaptic currents ($I_{Syn}$) and hyperpolarization-activated inward currents ($I_H$) were added via the dynamic clamp, following the methods described in *Sharp et al., 1996*; *Figure 1*. Simulation of voltage-dependent currents in real

time was done using Real-Time eXperimental Interface (RTXI 2.2 or 1.4) (http://rtxi.org/) (**Patel et al., 2017**). Custom RTXI modules were written using the programming language C++.

The synaptic current is given by the following expression:

$$I_{syn} = g_{syn} \cdot s \left(V_{pre}\right) \cdot \left(V_{post} - E_{syn}\right), \quad \left(1 - s_\infty\right) \frac{ds}{dt} = \frac{s_\infty - s}{\tau_{syn}},$$

where $V_{pre}$ and $V_{post}$ are presynaptic and postsynaptic voltages, $s$ is the synaptic gating varible, $s_\infty$ is the steady-state synaptic activation, given by a sigmoidal function $s_\infty = \frac{1}{1+e^{\frac{V-V_{th}}{V_{slope}}}}$ (**Figure 1B**, purple and orange curves).

The hyperpolariztion-activated inward current is described in **Buchholtz et al., 1992**:

$$I_H = g_H \cdot r \left(V_{post}\right) \cdot \left(V_{post} - E_H\right), \quad \frac{dr}{dt} = \frac{r_\infty - r}{\tau_H},$$

where $r$ is the gating varible of H current, $r_\infty$ is the steady-state activation, given by a sigmoidal function $r_\infty = \frac{1}{1+e^{\frac{V-V_{1/2}}{S_r}}}$ (**Figure 1B**, black curve), $\tau_r$ is the voltage-dependent time constant given by $\tau_H = \frac{\tau_{H0}}{1+e^{\frac{V-V_{\tau_r}}{S\tau_r}}}$.

In a subset of experiments, we simulated inward neuromodulatory current ($I_{MI}$) via dynamic clamp (**Swensen and Marder, 2001**):

$$I_{MI} = g_{MI} \cdot m \cdot \left(V_{post} - E_{MI}\right), \quad \frac{dm}{dt} = \frac{m_\infty - m}{\tau_m},$$

where $m$ is the gating varible of neuromodulatory current, $m_\infty$ is the steady-state activation, given by a sigmoidal function $m_\infty = \frac{1}{1+e^{\frac{V-V_{MI_{1/2}}}{S_{MI}}}}$.

**Table 1.** Parameter values for the dynamic clamp.

| Parameter | Value | Description |
|---|---|---|
| **Synaptic current ( $I_{syn}$)** | | |
| $g_{syn}$ | Varied from 150 to 1,050 nS | Maximal conductance of synaptic current |
| $E_{syn}$ | –80 mV | Reversal potential of synaptic current |
| $V_{th}$ | Varied from –28 to –54 mV | Synaptic threshold voltage |
| $\tau_{syn}$ | 50 or 100 msec | Synaptic time constant |
| $V_{slope}$ | –2 mV | Slope factor of synaptic activation function |
| **Hyperpolarization-activated inward current ( $I_H$)** | | |
| $g_H$ | Varied from 150 to 1,050 nS | Maximal conductance of H current |
| $E_H$ | –10 mV | Reversal potential of H current |
| $V_{1/2}$ | –50 mV | Half-maximal activation voltage of H current |
| $s_r$ | 7 mV | Slope factor of H current activation function |
| $\tau_{H0}$ | 2000 or 3000 msec | Time constant of H current |
| $V_{\tau_r}$ | –110 mV | Half-maximal voltage of H current time constant |
| $s_{\tau_r}$ | –13 mV | Slope factor of H current time constant |
| **Neuromodulatory inward current ( $I_{MI}$)** | | |
| $g_{MI}$ | 100, 150 or 200 nS | Maximal conductance of neuromodulatory current |
| $E_{MI}$ | –20 mV | Reversal potential of neuromodulatory current |
| $V_{MI_{1/2}}$ | –21 mV | Half-maximal activation voltage of $I_{MI}$ |
| $\tau_m$ | 4 msec | Time constant of neuromodulatory current |
| $s_{MI}$ | –8 mV | Slope factor of $I_{MI}$ activation function |

Parameter values of the currents injected in both neurons were the same to preserve the symmetry and are given in *Table 1*. Since the artificial currents injected into both neurons had the same parameter values, to create stable half-center oscillators, neurons used to comprised the oscillator had to have similar resting membrane potentials and intrinsic excitability. Thus, in the sunset of experiments, when the two GM neurons had very different resting membrane potentials at baseline, the membrane potential were brought to the same range of $\pm 5\,mV$ by either injecting a small amout of positive constant current or negative leak current to a more hyperpolarized cell.

## Temperature experiments

Temperature of the superfusing saline was controlled using either a waveform generator (RIGOL, DG1022 series) or Arduino connected to a temperature controller (model CL-100, Warner Instruments) and altered during each experiment using a Peltier device and thermocouple (SC-20 and TA-29, Warner Instruments). We performed three types of temperature experiments. In the first set of experiments the temperature was changed in one big step from 10°C to 20°C in 1 min, held at 20°C for 2–10 min and brought back to 10°C in one step (N = 13). In the second set of temperature experiments, the waveform generator or Arduino were programmed to change temperature from 10°C to 20°C in 2°C/min steps (N = 5). Each temperature step was held for 6 min during which synaptic threshold was changed via RTXI from –50 mV to –30 mV in 5 mV/min steps to explore the effect of temperature on half-center oscillator circuits with different oscillatory mechanisms. In the final set of temperature experiments, the waveform generator or Arduino was programmed to generate a smooth temperature ramp from 10°C to 20°C over 10–20 minutes (N = 22). Temperature was then held for 2–5 min at 20°C and gradually brought back to 10°C in a symmetric ramp. For a subset of temperature experiments (N = 15) inward neuromodulatory current I_MI was simulated via dynamic clamp in both GM neurons at either 10°C, 20°C or both temperatures.

For the escape mechanism, the synaptic thresholds were between –54 and -52 mV. Small variability in the synaptic thresholds comes from the variability in the resting membrane potentials of the neurons across preparations (*Figure 5C*). For the release mechanism, synaptic thresholds were between –44 and –32 mV. Variability in the synaptic thresholds for release mechanism comes from both the differences in the resting membrane potentials and the intrinsic excitability of the cells, such as spike threshold (*Figure 5D*) number of spikes per burst and spike frequency. Further, due to significant variability in the intrinsic excitability between the cells across the preparations, there was a significant variability in the maximal conductances of the synaptic and H currents across the experiments. The maximal conductances for the synaptic current and H currents varied between 200 and 900 nS across experiments.

Temperature dependence of the conductances and time constants of the currents generated with the dynamic clamp were implemented in the following way:

$$g_{syn} = g_{syn0} \cdot Q_{10}^{\frac{T-T_0}{10}}, \ g_H = g_{H0} \cdot Q_{10}^{\frac{T-T_0}{10}}, \ \tau_{syn} = \frac{\tau_{syn0}}{Q_{10}^{\frac{T-T_0}{10}}}, \ \tau_H = \frac{\tau_{H0}}{Q_{10}^{\frac{T-T_0}{10}}}$$

, where $T$ is the saline temperature and $T_0 = 10$ is a reference temperature. $Q_{10}$, a metric describing the rate of change of a biological process due to increase in temperature by 10°C, was set to either one or 2, according to experimentally measured $Q_{10}s$ in STG neurons (*Tang et al., 2010*).

## Quantification and statistical analysis

### Spike detection

Spikes were detected using local maxima detection algorithm in MATLAB, using a threshold of –40 mV and a peak prominence (height of the peak above the reference level) of 2. Prior to running local maxima algorithm voltage traces were smoothed using moving average filter with 10 data points for calculating smoothed value to reduce the noise in the traces.

### Burst detection

For an accurate detection of the bursts we used two methods: based on the spiking activity and based on the slow wave, as in most cases circuits exhibited prominent slow wave during alternating bursting.

## Burst detection based on the spiking activity

Bursts were identified as discrete events consisting of a sequence of spikes with burst onset defined by two consecutive spikes within an interval less than mean interspike interval ($ISI$) in a trace with set parameters, and burst termination defined by an $ISI$ greater than $ISI + 300\ msec$. Duty cycle ($DC$) was calculated as the burst duration divided by the cycle period (**Figure 2B**). Spike frequency was calculated as mean frequency of spikes within bursts.

## Burst detection based on the slow wave

Traces were low pass filtered to 1 Hz and smoothed using moving average filter with 100 data point windows. Then slow-wave peaks of membrane potential oscillations were detected using local maxima detection algorithm, with a threshold of mean value of filtered membrane potential ($V_{M_{filtered}}$) + 2.5 mV and a peak prominence of 3. Slow-wave dips were detected using the same algorithm for the inverted filtered traces. Slow-wave amplitude of membrane potential oscillation were calculated as the difference between peak and dip values. Cycle frequency of bursting circuits was calculated as an inverse of oscillation period determined by thresholding the filtered traces. Threshold was set to half the amplitude of the slow wave.

We manually inspected the traces to ensure the accuracy of burst and spike identification.

## Classification of a circuit activity patterns

Similar to **Grashow et al., 2009**, we classified the activity patterns of reciprocally inhibitory circuits into silent, asymmetric, irregular spiking and antiphase bursting (or half-center oscillations). To refine classification, we also added a 5th category, antiphase spiking (**Figure 4—figure supplement 1**).

Activity pattern was classified as silent if both neurons fired less than 5 spikes in 1 min. If only one of the cells fired more than 5 spikes in 1 min, the activity pattern was classified as asymmetric. If both cells were spiking, the pattern was classified as either irregular spiking, antiphase spiking or bursting. To distinguish these activity patterns, we calculated a measurement of burst exclusion, $\chi_{network}$, described in **Grashow et al., 2009**. This measure ranges from –1 (simultaneous bursts) to +1 (alternating bursts).

We determined active time intervals for each cell: if the neurons were bursting, the active time intervals were defined as the time from the first to the last spike in the burst. Otherwise the active time intervals were defined as ¼ the average interspike interval and centered on each spike. We then calculated the total active time for each cell, $t_{cell1}$ and $t_{cell2}$ as a sum of the active times of each respective cell, and the overlap time (when both cells were active) for the circuit, $O_{network}$. We then compared $O_{network}$ to the overlap times that would be expected for uncorrelated circuits, $O_{random}$, and the minimum possible overlap time, $O_{min}$.

$$\mathrm{O}_{min} = \left\{ \begin{array}{l} T_{trial} - t_{cell1} - t_{cell2} \quad if\ t_{cell1} + t_{cell2} > T_{trial} \\ \qquad\qquad 0 \quad otherwise \end{array} \right\}$$

$$\mathrm{O}_{random} = \left\{ \begin{array}{l} \min\left(t_{cell1}, t_{cell2}\right) - \frac{1}{2}\left[T_{trial} - max\left(t_{cell1}, t_{cell2}\right)\right] \quad if\ t_{cell1} + t_{cell2} > T_{trial} \\ \qquad\qquad \frac{\min\left(t_{cell1}, t_{cell2}\right)^2}{2\left[T_{trial} - max\left(t_{cell1}, t_{cell2}\right)\right]} \quad otherwise \end{array} \right\}$$

$T_{trial}$ is the total active time of the network, calculated as $T_{trial} = t_{cell1} + t_{cell2} - O_{network}$.
From this, we calculated the exclusion factor $\chi_{network}$ as

$$\chi_{network} = \frac{O_{random} - O_{network}}{O_{random} - O_{min}}$$

Circuits with both active cells were categorized as antiphase bursters (or half-center oscillators) if $\chi_{network} \geq 0.1$ and were characterized as spiking otherwise.

Finally, to determine whether the network exhibited antiphase spiking activity, we calculated percent of single spikes in bursts. If the percent of single spikes in bursts was more than 80%, we characterized the activity pattern of these circuits as antiphase spiking.

## Spectral analysis (Figures 5, 6, Figure 7-figure supplement 1)

Spectrograms for the temperature experiments were calculated using the *Burg, 1967* method for estimation of the power spectral density in each time window. The Burg method fits the autoregressive (AR) model of a specified order $p$ in the time series by minimizing the sum of squares of the residuals. The fast-Fourier transform (FFT) spectrum is estimated using the previously calculated AR coefficients. This method is characterized by higher resolution in the frequency domain than traditional FFT spectral analysis, especially for a relatively short time window (*Buttkus, 2000*). We used the following parameters for the spectral estimation: data window of 3.2 s, 50% overlap to calculate the spectrogram, and number of estimated AR coefficients $p$ = (window/4) + 1. Before calculating the power spectrum, voltage traces were low-pass filtered at 2 Hz using a six-order Butterworth filter and down-sampled.

## Statistics

To determine whether the duty cycle and spike frequency significantly increased/decreased with ERQ respectively, we measured the Spearman rank correlation coefficient ($\rho$) between the mean values of these characteristic and ERQ (*Figure 2D*). The Spearman correlation coefficient measures the strength and direction of correlation between two variables. $\rho = 1$ indicates that the two variables are a perfect monotonic function of each other.

To determine the $g_H$-$g_{Syn}$ conductances sets that produce statistically similar characteristics of the output of half-centers with escape and release mechanisms we performed Wilcoxon rank-sum test for each set of $g_H$-$g_{Syn}$ conductances (*Figure 3B–F*). Significance level was set to 0.05. The conductance sets producing the circuit output characteristics that were not significantly different ($p > 0.05$) are indicated by the red boxes in *Figure 3*.

To determine whether the GM neurons' resting membrane potentials, spike amplitudes and input resistances were significantly different at 10°C and 20°C we performed paired-sample Wilcoxon signed rank test (*Figure 5C and D*). To calculate the slopes of f-I curves we used a near-linear portion of the f-I curves at low injected currents. To determine whether the slopes are significantly different at 10°C and 20°C we performed paired-sample Wilcoxon signed rank test. The results of the statistical test can be found in the text and the legend of *Figure 5*.

To determine whether the Coefficients of Variation of cycle and spike frequencies in escape and release circuits were different at 10°C and 20°C we performed paired-sample Wilcoxon signed rank test (*Figure 5—figure supplement 1*).

To determine whether the characteristics of the output of half-centers with different oscillatory mechanisms and $Q_{10}$s were significantly different between 10°C and 20°C we performed paired-sample Wilcoxon singed rank test (*Figure 6F*). Significance level was set to 0.05. The results of the statistical test can be found in *Supplementary file 1b* and in the text. To determine whether the changes in characteristics with an increase in temperature were significantly different between the circuits with release and escape mechanisms and different temperature-dependences we performed one-way ANOVA with Tuckey post-hoc using IBM SPSS Statistics 24. The results of one-way ANOVA can be found in *Supplementary file 1c-1h*.

To determine whether the characteristics of the circuit output were significantly different after the addition of the neuromodulatory current we performed paired-sample t-test (*Figure 8D1-5*). Significance level was set to 0.05. The results of the statistical test can be found in the legend of *Figure 8*.

To determine whether the Coefficient of Variation of cycle frequency of the circuits operating with different mechanisms was significantly different in control and with addition of the neuromodulatory current, we performed paired-sample Wilcoxon signed rank test (*Figure 8—figure supplement 1*). Significance level was set to 0.05.

## Data and code availability

Data have been deposited at Zenodo and is publically available at https://zenodo.org/record/5504612 (DOI: 10.5281/zenodo.5504612).

Custom RTXI modules are available on GitHub (https://github.com/eomorozova/half_center_oscillator_rtxi_module, copy archived at swh:1:rev:d42be99960f2a73057a14483ed051ec326b96fcb, *Morozova, 2022a*).

All the analysis scripts are available on GitHub (https://github.com/eomorozova/hco-analysis, copy archived at swh:1:rev:2df3ea4a3cfdfb98bb2740655239c5e0dc3e1dd5, *Morozova, 2022b*).

Any additional information required to reanalyze the data should be directed to Ekaterina Morozova (morozova.e.o@gmail.com).

## Acknowledgements

Support: NIH National Institute of Health grant 2 R01 MH046742, and the Swartz Foundation (EOM).

## Additional information

### Funding

| Funder | Grant reference number | Author |
| --- | --- | --- |
| National Institutes of Health | 2 R01 MH046742 | Eve Marder |
| Swartz Foundation | | Ekaterina Morozova |

The funders had no role in study design, data collection and interpretation, or the decision to submit the work for publication.

### Author contributions

Ekaterina Morozova, Conceptualization, Data curation, Formal analysis, Investigation, Methodology, Project administration, Software, Validation, Visualization, Writing - original draft, Writing – review and editing; Peter Newstein, Data curation, Formal analysis, Investigation, Project administration, Writing – review and editing; Eve Marder, Conceptualization, Funding acquisition, Methodology, Resources, Supervision, Writing – review and editing

### Author ORCIDs

Ekaterina Morozova ![ORCID] http://orcid.org/0000-0001-9131-7756
Peter Newstein ![ORCID] http://orcid.org/0000-0003-2966-783X
Eve Marder ![ORCID] http://orcid.org/0000-0001-9632-5448

### Decision letter and Author response

Decision letter https://doi.org/10.7554/eLife.74363.sa1
Author response https://doi.org/10.7554/eLife.74363.sa2

## Additional files

### Supplementary files

• Supplementary file 1. Summary statistics for the temperature experiments. (a) Mean ± SD of output characteristics of the circuits in escape and release at 10°C and 20°C. (b) Significance analysis of the cycle frequency, spike frequency, number of spikes per burst, slow wave amplitude, duty cycle and ERQ at 10°C and 20°C. (c-h) Significance analysis of the change in the output characteristics of the circuits in escape and release with different temperature-dependencies.

• Transparent reporting form

### Data availability

Data as been deposited at Zenodo (https://doi.org/10.5281/zenodo.5504612).

The following dataset was generated:

| Author(s) | Year | Dataset title | Dataset URL | Database and Identifier |
|---|---|---|---|---|
| Ekaterina OM, Peter N, Eve M | 2022 | Reciprocally inhibitory circuits with distinct mechanisms are differentially robust to perturbation and modulation | https://doi.org/10.5281/zenodo.5504612 | Zenodo, 10.5281/zenodo.5504612 |

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
