## [Editor Report]

Morozova et al., describe potential mechanisms contributing to the flexibility of burst patterns and dynamic responses to perturbations within an isolated reciprocally inhibitory circuit derived from the stomatogastric ganglion of the crab. The authors use the dynamic clamp approach to study the interactions between pharmacologically isolated, intrinsically silent gastric mill neurons. The authors demonstrate that the mechanisms of oscillation of the half-center networks are not fixed and shift to favor a release or escape mechanism depending on the synaptic threshold, IH conductance, and synaptic conductance. They also show that the different mechanisms of oscillation are differentially sensitive to neuromodulation and temperature changes. This is a fundamentally important study because reciprocally organized networks are ubiquitous and found virtually in every organism.

---

## [Decision Letter]

**Decision letter after peer review:**

Thank you for submitting your article "Reciprocally inhibitory circuits operating with distinct mechanisms are differently robust to perturbation and modulation" for consideration by *eLife*. Your article has been reviewed by 3 peer reviewers, and the evaluation has been overseen by Ronald Calabrese as the Senior Editor and Reviewer #1. The following individuals involved in review of your submission have agreed to reveal their identity: Paul Katz (Reviewer #2); Jan Marino Ramirez (Reviewer #3).

Essential revisions:

1. Address concerns about the operational definition of robustness in the paper.

2. Clarify with new analyses questions about rhythm variability.

3. If conclusions are based on single or a few exemplars then either support them more fully with analyses of pooled data or scale back on the conclusions.

*Reviewer #1:*

This is a very careful and systematic hybrid system analysis of the mechanism underlying alternating bursting in mutually inhibitory neurons or half-center oscillators (HCOs). By clever use of dynamic clamp, the authors create HCOs between non-connected living neurons (from the crab stomatogastric ganglion) of the same type by adding artificial synapses and h-current. This hybrid system allows them to manipulate synaptic threshold as a control variable to engage different oscillatory mechanisms escape and release, which are based on a theoretical understanding of HCO operation. They also have control of synaptic and h-current conductance and dynamics (activation-deactivation) and manipulate these variables or as proxies for changes in temperature of circuit operation. Using the synaptic threshold control variable to set escape or release mode they discern difference of these manipulations on burst characteristics in escape vs release modes. In separate experiments, they also add a modulatory current (similar to a persistent Na current) in dynamic clamp and explore it effects on HCOs in escape and release modes. The end result is a thorough analysis of how oscillator mechanism in an HCO, a basic circuit building block, affects circuit responses to perturbation and modulation.

The experiments are well performed, and a deep and rich data set is generated that is appropriately analyzed. The findings are significant for all interested in oscillatory network function and its resilience to perturbation and modulation.

Concerns:

Robustness is often mentioned but is not precisely defined. Operationally robustness seems in this paper to stand for robustness to (1) activity regime change under parameter variation, (2) stability of burst characteristics with parameter variation, and (3) slow-wave amplitude, spiking strength (spike frequency), and symmetry of bursting. These are three very different things and should be clearly differentiated in the text so that when robustness is mentioned, the type of robustness is made clear. Perhaps robustness should be limited to the first, activity regime, and some other terms used for the other two.

On several occasion in the text the authors refer to irregularity in bursting of the hybrid HCOs, but this is not quantified beyond displaying exemplars that seem to have irregular bursting. Pooled data should be analyzed in the different modes and manipulations and analyzed for statistical difference in the CoV of cycle frequency (or period) and burst duration. Similarly, the authors cite changes in symmetry in bursting in exemplars but do not present pooled quantitative data in support of the claim, just visual inspection of exemplars.

In the stomatogastric networks, synaptic transmission is largely graded (based on release mediated by the slow wave of oscillation) and not so much spike-mediated, so it is reasonable that synaptic threshold should be a control variable in this system. Moreover, spikes, recorded in the cell bodies are not reflective of their amplitude at the SIZ. In other system transmission can be largely mediated by spikes. At the beginning of the paper (Figure 1), it is clear that release mode in their hybrid HCOs depends on spike-mediated transmission because synaptic threshold is above the slow-wave depolarization, thus spike frequency is a key feature determining the mechanism of oscillation. However, in escape mode the transmission is purely graded because synaptic threshold is so low that transmission is saturated by the slow-wave depolarization and spikes contribute little if anything, thus spike frequency is immaterial to the mechanism of oscillation. This situation should be addressed at the beginning of the paper in reference to Figure 1. How this spike-mediated vs. graded balance plays out in the mixed mechanism modes remains to be explored.

In Figure 1C, the authors show convincingly that there is a vast landscape where their hybrid HCO operate in a mixed mechanistic mode somewhere between escape and release corresponding to synaptic thresholds in the middle range. This mixed mode is addressed only with a single exemplar in Figure 8B as a case for how modulation affects mixed mode circuits. The Discussion should reflect plainly that this mixed mode is likely common in biological circuits and may go hand-in-hand with significant reliance on spike-mediated transmission.

The authors state "The modulatory current (IMI) restores oscillations in release circuits but has little effect in escape circuits." but this is supported by a single exemplar (Figure 8E) and no pooled data is presented.

1. Lines 130-131: make clear that h-current must also be added by dynamic clamp.

2. Line 158: the definition of VM-bar is not clear. I was confused by this in Figure 1B. My expectation from Methods is that with proper filtering at 1Hz that spikes would be eliminated (Does your filter integrate? This would seem problematic for calculating ERQ. What is the filter? Simple RC?) and then the smoothing would leave a slow wave that peaks near the dashed red line, so VM-bar would be slightly below the dashed red line. Please indicate VM-bar on each exemplar trace of Figure 1A and rationalize VM-bar determination explicitly.

3. Line 203: In the exemplar records shown in Figure 2 A, GM2 has a very low spike frequency in the escape mode and higher in mixed.

4. As synaptic threshold becomes more positive the importance of spike-mediated transmission appears to become more important. Are all releases in this system using purely spike mediate transmission as suggested by Figure 2 A and C?

5. Lines 220-223: Can you state explicitly what criteria you applied to determine to designate escape vs. release (ERQ?) for analysis, and especially how synaptic threshold varied among the HCOs within each group? This is especially important given this last sentence of this paragraph.

6. Lines 228-231: Can you provide data in support of these conclusions? Quantify h-current at the point of transition, for example? If such data are not available, these conclusions could be scaled back a bit.

7. Lines 236-238: If I understand Methods correctly, your definition of duty cycle is not appropriate for most or all of the designated escape circuits. In escape and near escape, synaptic transmission occurs while the cell still depolarized above synaptic threshold but is not spiking. Better to define duty cycle as time above threshold/cycle period; you are discussing the rhythm generating function here not the motor outflow.

8. Line 236: Figure 4A and B?

9. Lines 263-271: I am very confused by this paragraph. Remind the reader of the ERQ criteria for escape and release. (a) Are the extremes i and v escape and release both mixed?? "The ERQ threshold for escape is −0.038 {plus minus} 0.008, while the ERQ threshold for release is 0.105 {plus minus} 0.012." It is hard to tell from the color code of Figure 2A whether these criteria are ever met. (b) How is the mechanism changing? Can you state that explicitly? I interpret that the extremes of your HCO hybrid system as being escape and release with reality sitting in the middle, which is mixed. Are you postulating distinct mixed mechanisms? Can you define them? (c) I think that you are saying that the balance between release promoting ionic currents and escape promoting ionic currents are changing but you do not define these currents. In the mixed regime, one should speak about ionic currents if one is to speak of mechanism.

10. Lines 289-291: What about systems that rely on spike-mediated transmission?

11. Line 294: In what sense is bursting less regular? Can you define the criteria and present numerical data? Maybe Coefficient of Variation of the cycle period or frequency?

12. Lines 296-301: Can you provide evidence for these conclusions by analyzing each cell separately for ERQ? Are you saying that in mixed mode that one cell is escaping, and one is being released? What exactly are you thinking here?

13. Line 312: STAR methods?

14: Lines 320-323: Asymmetry in activity does not require asymmetry in excitability. You could have bistability or a (possibly even artifactual) difference in baseline membrane potential despite your attempts to equalize them with injected current. This statement requires more evidence to be firm.

15. Lines 327-341: What values were chosen for gh and gSyn in each case (escape vs release)? Are these the same throughout all experiments???? How about Vth, is this the same across experiments for each case?

16. Line 344: Can you quantify this irregularity as CoV? I see an exemplar that seems more variable but also a lot faster (higher frequency). Are there pooled data to support claims on cycle frequency and its variability?

17. Lines 346-348: Is this surprising? The exemplar is dependent solely on spike-mediated transmission and so is susceptible to changes in spiking.

18. Lines 385-388: This seems an understatement. This system at elevated Vth is operating in spike-mediated transmission mode and so spikes and spike frequency are all important.

19. Lines 394-395: I don't see this in the data. I see a distinct crossover at around 4 nA – 15Hz. Can you quantify slopes? Is there pooled data to support this conclusion?

20. Lines 402-411: You are here recognizing the distinct difference between HCOs that operate with graded vs spike-mediated inhibition. This should be thoroughly aired in relation to Figure 1; don't wait till here.

21. Lines 411-413: Can you support this conclusion with data?

22. Lines 418-421: There are real difference between a slow wave HCO intrinsic excitability and a spiking HCO. For one a weakly regenerative current like a INaP can lead to weak rebound, but robust spiking supporting an HCO based on spikes. I suspect in the case Figure 5B1 that you either have a very long lingering h-current or a LT relatively slowly inactivating Ca current that is fully inactivated when Vth is more positive and inhibition weaker. You do not measure rebound in the GM neurons.

23. Figure 6: I find the logic of this figure unclear. The purpose is to study the effect of temperature on the hybrid HCOs in different oscillation modes. So, you present HCOs with the added h and synaptic conductances at standard temperature for a comparison group (Panel A), but then Panel B is simply to illustrate changes associated with changes in h and synaptic conductances at a different temperature, and finally in Panel C you explore a simulated change in temperature. I understand why you have Panel B included (to parse mechanism) but shouldn't the order be ACB?

24. Lines 451-453: Do you have data to support this conclusion?

25. Line 456: '…temperature-independent synapse…' This is getting confusing. It is temperature insensitive h-current and synapses; please make it clear.

26. Line 460: '…release or escape…'

27. Lines 479-480: Here it would be good to remind the reader that the deactivation rate is as important as the activation rate and that in fact they are two sides of the same coin. From Methods I think you just changed Tau0s (2X or 1.5X) (τ*H*0 2000 or 3000 msec, τ*s*yn 50 or 100 msec) with temperature so both activation and deactivation are affected. I suggest that you plot the Taus of the synapses and the h-current on the plots of Figure 1B. You do provide the equations in Methods, but I think a visualization would help.

28. Lines 486-489: Any data to support these conclusions? Change activations rates without affecting deactivation?

29. Lines 491-492: Can you provide more evidence in support here? I would like to know what is causing the rebound in the GM neurons before I can fully accept this conclusion. If the rebound is due to lingering h, maybe emphasize the deactivation.

30. Lines 518-521: Please put the numbers here in the text, at least for the Q10 g and k case. The Table is pretty tough to isolate on.

31. Lines 521-525, Figure S2: This exemplar in Figure S2 is not convincingly escape at 10 C. Pure escape should transition to the depolarized state when threshold is reached. This is a good place for me to comment on the ERQ criterion for designating escape and release. I appreciate the need for an automatable algorithm to designate escape vs. release but I have two caveats. (1) Classically escape is designated by transition when the inhibited cell reaches synaptic threshold (or maybe in spike-mediated transmission spiking threshold) and release is designated by transition when the depolarized cell crosses the synaptic threshold (or stops spiking). The case in Figure S2 clearly violates this classical designation. (2) ERQ is determined by averaging across the two neurons in the HCO, this hides asymmetries and allows for the two cells to have different mechanism, e.g., one escape and one mixed. ERQs must be calculated independently for the two neurons if you wish to reveal asymmetries.

32. Lines 547-550: Only one prep is illustrated in Figure 8C. Is the CoV of bursting in mixed mode different across pooled data to support this statement?

33. Line 590: '…vastly…'

34. Lines 593-597: Only one prep is illustrated in Figure 8E. Are there pooled data that can support this conclusion?

35. Lines 650-653: Because asymmetry is not directly assessed in this paper this conclusion should be scaled back.

36. Lines 653-656: Only one prep is assessed in mixed mode, so this statement should be scaled back.

37. Lines 687-689: Please be careful here and designate modes precisely and state exactly what you mean by stable cycle frequency. Does this mean regularity of bursting or period constancy? Ditto for living preparations with intact networks. Line 693: does temperature compensation in intact STG networks involve a constant period of just constant phase?

38. Lines 626-628: Are there modeling studies involving more realistic neurons specifically ones that spike and use spike-mediated transmission.

*Reviewer #2:*

This manuscript provides a very detailed and thorough examination of an important issue in neural circuit research, namely how the mechanisms underlying neural activity relate to robustness in the face of perturbations. It examines the simplest neural circuit possible, one involving just two neurons that reciprocally inhibit each other, which is capable of producing rhythmic alternating activity. The research shows that there is a continuum of mechanisms based on synaptic and membrane properties of the two neurons that can generate a robust output. At one end of the continuum, each neuron escapes from the inhibition of the other. At the other end, each neuron releases the other from inhibition. In the middle, both mechanisms contribute to generation of rhythmic activity. The effects that perturbations such as temperature and neuromodulators have on the circuit depend upon where the mechanism of oscillation lies along this continuum.

This paper has several important strengths:

It uses dynamic clamp technique to artificially couple two real neurons and provide them with a membrane conductance that they don't normally have. This is a powerful technique that merges experimental and theoretical neuroscience because the researchers are able to systematically alter parameter values such as synaptic strength and ionic conductance that are not feasible to modify biologically. Yet they are also monitoring the activity of real neurons.

The manuscript thoroughly represents the results and convincingly demonstrates how release and escape mechanisms are differentially affected by perturbations. The method of data visualization is very effect at summarizing complex results.

An important conclusion drawn from the results is that half-center oscillators using a release mechanism are more robust to variations in synaptic and membrane conductance.

Another important conclusion is that the same circuit can produce a similar output using different mechanisms and that it is not possible to know which mechanism is used without looking at the effect of perturbation.

I would encourage the authors to not start the abstract with a question, but rather use a standard topic sentence that gets right to the problem.

The introduction could be firmed up more. For example, the sentence "Lateral inhibition is important in many sensory systems, and reciprocal inhibition between individual neurons or groups of neurons is the 'building block' of many half-center oscillators that generate antiphase and multiphase activity patterns." Is a run-on referring to lateral inhibition in sensory systems and then going into detail about reciprocal inhibition in HCOs. The concluding sentence of the introductory paragraph does not follow from the content of the paragraph.

Figure 3A. If I understand this correctly, 10 models were made with each of the 49 combinations of gH and gSyn. The percentages then are going to limited to 0,10,20,30,40…100%. To make that clear, change the continuous gray scale "%oscillators" to the 10 discrete gray values (as was done in Figure S1A). That will provide the reader with more information about the values. Same with Figure 3F.

Figure 3C, the red boxes are nearly invisible and I imagine that Figure 3D is not color-blind friendly.

Figure S1A, I found the colors difficult to distinguish. The trends did not pop out at me.

Figure 7 refers to 'cases' that are also in Figure 6, but not referred to as cases. It took me a while to recognize that they were the same. It would be helpful to label case 1, case 2, and case 3 in Figure 6 A1, B1, C1 and in Figure 7 A-D in the Figure It would also help to refer the cases in the text for consistency such as line 438, Case 2 and line 475 Case 3.

Also, recognizing color equivalence in Figure 7 A-F lines and E-H boxes is really hard. I think it's because the box plots have a contrast-enhancing black border. I'm starting to think that I may be color-impaired.

Line 546-550: "I-MI made oscillations less stable and irregular for circuits operating with a mixture of mechanisms." Stating "This is obvious…" is not an explanation; although it may be obvious to the authors, it needs to be explained to the readers.

Regarding the 2nd point, "an increase in the standard deviation of the cycle frequency", I don't see documentation of this; the error bars in Figure 8C are larger at the release end of the graph than in the middle.

*Reviewer #3:*

The authors demonstrate that the mechanisms of switching between components of the reciprocally organized half-center network are not fixed and may shift to favor a release or escape mechanism depending on factors such as the synaptic threshold, Ih conductance, and synaptic conductance. This is a fundamentally important study because reciprocally organized networks are ubiquitous and found virtually in every organism.

This study leads to the important conclusion that a given rhythmic output alone does not reveal the underlying rhythmogenic mechanisms. A rhythmic output is not based on one "fixed" mechanism, but on the interplay between different rhythmogenic modes. Moreover, because of this interplay it is impossible to predict how this network will respond to perturbations.

The study is an important reminder that even a small two neuron network with a well defined, extremely simple "connectome" is strikingly flexible and complex: an important lessons for those aspiring to obtain complete connectomes in mammals in the hope to reveal the secrets of the brain.

The authors use the dynamic clamp approach to study the interactions between pharmacologically isolated, intrinsically silent gastric mill neurons, an approach pioneered by Andrew Sharp in the 1990's. Because of individual differences in the intrinsic properties from neuron to neuron, which is very characteristic for numerous networks, the authors introduce the escape to release quotient (ERQ) to be able to pool the responses of different neurons and demonstrate that changing the synaptic threshold can transform the network in a sigmoid manner from one dependent on an escape to one relying on the release mechanisms. Additionally, the authors demonstrate a network favoring a release mediated mechanism of switching responds differently to perturbation and modulation in H-current and synaptic conductances, compared to a network favoring an escape mechanism, despite similar patterns at rest. This is a fascinating finding, since a given rhythmic output alone does not reveal whether it is favoring one or the other switching mechanism and therefore also does not reveal how it will respond to perturbation. The differences can be striking: increasing an Ih current can lead to an increase or decrease in frequency, which could explain why blockade of the Ih current may yield inconsistent results. Similarly, bursting can be more or less regular dependent on the synaptic threshold. Overall, the manuscript is very comprehensive, and elegantly mechanistic. Because the dynamic clamp approach allows the investigators to carefully dissect the contributions of each of these cellular parameters it serves as a fundamental framework for understanding rhythmogenesis in general, which has always been the strength of the stomatogastric ganglion. Additionally, the difficulty in performing these elegant studies should be commended. It is striking how flexible and complex a simple two-neuron network can be: take that for those believing that a complete connectome will reveal the secrets of the brain. But it is also an important reminder that it is impossible to explain network functions based on characterizing firing patterns alone.

---

## [Author Response]

Essential revisions:1. Address concerns about the operational definition of robustness in the paper.

Defining robustness is nontrivial. We added a full paragraph to the Introduction (lines 103-114) describing how we define circuit robustness and challenges associated with establishing which features are central to robustness. We revisited instances in the paper, in which we refer to robustness, and clarified whenever possible whether we are talking about the change in the qualitative state of the circuit and/or sensitivity of certain features of the circuit output that might bring the circuit closer to the transition to another qualitative state.

2. Clarify with new analyses questions about rhythm variability.

We performed additional analyses to quantify irregularity of the circuit rhythms at baseline, during temperature changes and with the addition of a neuromodulatory current. We calculated the Coefficient of Variation (CV) of the cycle frequency and spike frequency of the circuits in release and escape at low and high temperatures. We also quantified irregularity and asymmetry of circuits operating with a mixture of mechanisms in control and in the presence of a neuromodulatory current, by calculating the CV of cycle frequency and a difference in burst durations between neurons. These data are included as supplements to Figures 4, 5 and 8.

3. If conclusions are based on single or a few exemplars then either support them more fully with analyses of pooled data or scale back on the conclusions.

We supported these conclusions by performing additional analyses of the pooled data and by providing more examples to support our claims. These data are included in the text and in the supplementary figures.

Reviewer #1:This is a very careful and systematic hybrid system analysis of the mechanism underlying alternating bursting in mutually inhibitory neurons or half-center oscillators (HCOs). By clever use of dynamic clamp, the authors create HCOs between non-connected living neurons (from the crab stomatogastric ganglion) of the same type by adding artificial synapses and h-current. This hybrid system allows them to manipulate synaptic threshold as a control variable to engage different oscillatory mechanisms escape and release, which are based on a theoretical understanding of HCO operation. They also have control of synaptic and h-current conductance and dynamics (activation-deactivation) and manipulate these variables or as proxies for changes in temperature of circuit operation. Using the synaptic threshold control variable to set escape or release mode they discern difference of these manipulations on burst characteristics in escape vs release modes. In separate experiments, they also add a modulatory current (similar to a persistent Na current) in dynamic clamp and explore it effects on HCOs in escape and release modes. The end result is a thorough analysis of how oscillator mechanism in an HCO, a basic circuit building block, affects circuit responses to perturbation and modulation.The experiments are well performed, and a deep and rich data set is generated that is appropriately analyzed. The findings are significant for all interested in oscillatory network function and its resilience to perturbation and modulation.Concerns:Robustness is often mentioned but is not precisely defined. Operationally robustness seems in this paper to stand for robustness to (1) activity regime change under parameter variation, (2) stability of burst characteristics with parameter variation, and (3) slow-wave amplitude, spiking strength (spike frequency), and symmetry of bursting. These are three very different things and should be clearly differentiated in the text so that when robustness is mentioned, the type of robustness is made clear. Perhaps robustness should be limited to the first, activity regime, and some other terms used for the other two.

We added a full paragraph to the Introduction (lines 103-114) describing how we define circuit robustness and challenges associated with establishing which features are central to robustness. We revisited instances in the paper, in which we refer to robustness, and clarified whether we are talking about the change in the qualitative state of the circuit and/or sensitivity of certain features of the circuit output that might bring the circuit closer to the transition to another qualitative state.

On several occasion in the text the authors refer to irregularity in bursting of the hybrid HCOs, but this is not quantified beyond displaying exemplars that seem to have irregular bursting. Pooled data should be analyzed in the different modes and manipulations and analyzed for statistical difference in the CoV of cycle frequency (or period) and burst duration. Similarly, the authors cite changes in symmetry in bursting in exemplars but do not present pooled quantitative data in support of the claim, just visual inspection of exemplars.

As suggested, we analyzed pooled data for irregularity and asymmetry in different modes and conditions and presented these data in supplementary figures to Figures 4, 5 and 8. Particularly, we quantified the irregularity of the rhythms by calculating the CV of cycle frequency of the circuits operating with different mechanisms (Figure 4 —figure supplement 1A). We calculated the CV of cycle and spike frequencies of escape and release circuits at different temperatures (Figure 5 —figure supplement 1). Finally, we calculated the CV of cycle frequency of circuits operating with a mixture of mechanisms in control and with addition of the neuromodulatory current (Figure 8 —figure supplement 1A).

To quantify the asymmetry in bursting in different conditions, we calculated the difference in the burst durations between neurons in the circuits with different synaptic thresholds (Figure 4 —figure supplement 1B), ERQ values for each neuron independently (Figure 4 C,D, Figure 7 —figure supplement 1), and the difference in the number of spikes per burst between neurons in a circuit in control and with the addition of I_MI_ (Figure 8 —figure supplement 1B).

In the stomatogastric networks, synaptic transmission is largely graded (based on release mediated by the slow wave of oscillation) and not so much spike-mediated, so it is reasonable that synaptic threshold should be a control variable in this system. Moreover, spikes, recorded in the cell bodies are not reflective of their amplitude at the SIZ. In other system transmission can be largely mediated by spikes. At the beginning of the paper (Figure 1), it is clear that release mode in their hybrid HCOs depends on spike-mediated transmission because synaptic threshold is above the slow-wave depolarization, thus spike frequency is a key feature determining the mechanism of oscillation. However, in escape mode the transmission is purely graded because synaptic threshold is so low that transmission is saturated by the slow-wave depolarization and spikes contribute little if anything, thus spike frequency is immaterial to the mechanism of oscillation. This situation should be addressed at the beginning of the paper in reference to Figure 1. How this spike-mediated vs. graded balance plays out in the mixed mechanism modes remains to be explored.

We added a description of graded vs spike-mediated transmission in escape and release modes to the beginning of the Results section (lines 173-179).

In Figure 1C, the authors show convincingly that there is a vast landscape where their hybrid HCO operate in a mixed mechanistic mode somewhere between escape and release corresponding to synaptic thresholds in the middle range. This mixed mode is addressed only with a single exemplar in Figure 8B as a case for how modulation affects mixed mode circuits. The Discussion should reflect plainly that this mixed mode is likely common in biological circuits and may go hand-in-hand with significant reliance on spike-mediated transmission.

We added a paragraph to the Discussion section reflecting that a mixture of mechanisms is common in biological systems and discussing that there is a continuum of mechanisms that can exist in rhythmic circuits (lines 722-743). We show in the paper that the balance of the mechanistic operations is sensitive to parameter variations and perturbations and can be biased towards one or the other mechanism on the vast landscape between escape and release.

In this paper we mostly focused on describing the behavior of the system operating at the extremes of this continuum, the synaptic escape and release mechanisms, because they are more identifiable mechanisms. However, we do describe the properties of the circuits operating in mixed modes and the transition in the mechanisms at multiple instances. Figure 2 shows how characteristics of the circuits change as they transition through the mixtures of mechanisms. Figure 4 shows how output characteristics of the circuits operating in a mixed mode depend on the changes in conductances. We also added the analysis of the pooled data from the circuits operating in the mixture of mechanisms in the presence of I_MI_ and added these data to the supplement of Figure 8.

The authors state "The modulatory current (IMI) restores oscillations in release circuits but has little effect in escape circuits." but this is supported by a single exemplar (Figure 8E) and no pooled data is presented.

We performed 4 experiments, in which oscillations of the circuits with a release mechanism were lost at high temperature and restored by adding I_MI_ to both neurons. We added a statement to the text of the manuscript that the oscillations were restored in 4/4 circuits with the addition of I_MI_ (line 699). We have provided a single example trace in the paper, because the effect of I_MI_ was consistent across all the preparations. We provide additional examples of I_MI_ rescue in response to the question #35.

1. Lines 130-131: make clear that h-current must also be added by dynamic clamp.

We added a line saying that we added H current via dynamic clamp to generate alternating bursting pattern of activity.

2. Line 158: the definition of VM-bar is not clear. I was confused by this in Figure 1B. My expectation from Methods is that with proper filtering at 1Hz that spikes would be eliminated (Does your filter integrate? This would seem problematic for calculating ERQ. What is the filter? Simple RC?) and then the smoothing would leave a slow wave that peaks near the dashed red line, so VM-bar would be slightly below the dashed red line. Please indicate VM-bar on each exemplar trace of Figure 1A and rationalize VM-bar determination explicitly.

VM-bar stands for the mean membrane potential of the neurons. For the ERQ calculations, the traces are not filtered, because, as you mentioned, that would be problematic considering the fact that spikes significantly contribute to switching behavior for more depolarized synaptic thresholds.

We only apply the filter for burst detection, spectral analysis, and quantification of the slow wave amplitude. The filter we use is a six-order low-pass Butterworth filter as described in the spectral analysis section of the Methods and Materials.

We apologize for the confusion caused by using the same notation of for both raw and filtered traces, we changed the notation of the mean membrane potential of the filtered traces to VM-bar_filtered.

We indicated on each exemplar trace in Figure 2A with solid black lines.

3. Line 203: In the exemplar records shown in Figure 2 A, GM2 has a very low spike frequency in the escape mode and higher in mixed.

The exemplar trace has many features that represent typical behavior; however, the GM2 neuron has very few spikes per burst and as this comment suggests, a lower spike frequency in escape than in a mixed mode. The data shown in this trace also appears as a black line in Figure 2 D5, which shows the dependence of spike frequency on ERQ for each individual experiment. However, on average as shown by the red line, spike frequency within burst decreases as the mechanism of oscillations changes from escape to release ( Spearman rank correlation test).

4. As synaptic threshold becomes more positive the importance of spike-mediated transmission appears to become more important. Are all releases in this system using purely spike mediate transmission as suggested by Figure 2 A and C?

That is correct, the more depolarized the synaptic threshold, the larger the contribution of spikes to the synaptic current. We added a small paragraph to lines 173-179 describing this phenomenon. However, because the synaptic activation function is not a step function, but a steep sigmoid function, which more realistically describes synaptic transmission, the synaptic current starts to activate at the membrane potentials below the synaptic threshold. Skinner et al., 1994 investigated the dependence of the mechanism of oscillation on the steepness of the synaptic threshold. They found that if the oscillations are further from the relaxation type, the transition between the mechanisms becomes less sharp. That being said, at the ends of the continuum, where circuits operate with a synaptic escape or release mechanisms, the behavior of the biological circuits closely resembles the theoretical escape or release scenarios.

In Author response image 1 we are including an example plot showing synaptic activation in escape and release modes. In escape mode the synaptic activation function is saturated at the spikes, and the spikes have virtually no contribution to the synaptic current. In release, synaptic activation mostly relies on spikes, but there is also some contribution from the slow wave, which makes on-off transitions more robust.

**Author response image 1. sa2fig1:** 

5. Lines 220-223: Can you state explicitly what criteria you applied to determine to designate escape vs. release (ERQ?) for analysis, and especially how synaptic threshold varied among the HCOs within each group? This is especially important given this last sentence of this paragraph.

We added a paragraph describing how the synaptic thresholds varied among half-centers in escape and release and the ERQ criteria used for determining the mechanisms in each case (lines 254-263). Particularly, the mechanism of oscillation was determined based on the ERQ thresholds of -0.038 for escape and 0.105 for release that we calculated based on the experiments shown in Figure 2. The ERQ values corresponding to the same mechanism vary depending on the values of the maximal conductances. Threshold ERQ values of -0.038 for escape and 0.105 for release correspond to gH=300nS and gSyn=200nS, which typically are the lowest conductance combination resulting in stable oscillations. Higher values for conductances correspond to more negative ERQ values in case of escape and more positive ERQ values in case of release, meaning that the threshold values obtained for the lowest conductance combinations can be used for most networks examined in our study.

6. Lines 228-231: Can you provide data in support of these conclusions? Quantify h-current at the point of transition, for example? If such data are not available, these conclusions could be scaled back a bit.

To support our conclusion that an increase in H-conductance decreases the oscillation frequency in release by prolonging the active phase of the oscillations and increases the frequency in escape by helping the neuron cross the synaptic threshold faster, in Author response image 2 we are providing sample voltage traces in release and escape at two different values of H-conductance. The traces show a prolongation of the active phase of oscillation with the increase in H-conductance in release. In escape, increase in H-conductance leads to a faster depolarization of the inhibited neuron above the synaptic threshold. Maps on the right show the dependence of the frequency of oscillations on the maximal value of H-conductance in release and escape in a single preparation.

7. Lines 236-238: If I understand Methods correctly, your definition of duty cycle is not appropriate for most or all of the designated escape circuits. In escape and near escape, synaptic transmission occurs while the cell still depolarized above synaptic threshold but is not spiking. Better to define duty cycle as time above threshold/cycle period; you are discussing the rhythm generating function here not the motor outflow.As suggested, we calculate the duty cycle as time above the synaptic threshold. In the case of release, the original calculation is based on the duration of spiking holds, because the synaptic threshold is close to the top of the slow wave and spiking plays a major role in switching behavior. In the case of escape, we calculate the duty cycle based on the slow wave, in additional to the original analysis based on spikes. We are including this analysis as a supplementary figure to figure 3. The mean duty cycle of the circuits in escape, calculated based on the slow-wave, is .

8. Line 236: Figure 4A and B?

Do you mean line 263? We changed it to Figure 4A and B.

9. Lines 263-271: I am very confused by this paragraph. Remind the reader of the ERQ criteria for escape and release.

We added a sentence reminding the readers of the ERQ criteria we used to determine the mechanism of oscillation (lines 315-317).

a) Are the extremes i and v escape and release both mixed?? "The ERQ threshold for escape is −0.038 {plus minus} 0.008, while the ERQ threshold for release is 0.105 {plus minus} 0.012." It is hard to tell from the color code of Figure 2A whether these criteria are ever met.

In the experiment shown in Figure 4A, the circuit operates in a mixed mode in the upper-left region of the parameter space and release mode in the lower-right region. In the experiment shown in Figure 4C,D, the circuit spans the entire range of mechanisms from escape to release in the (gH, gSyn) parameter space corresponding to the synaptic threshold of -40 mV. We outlined the regions in the parameter space corresponding to different mechanisms of oscillation (release, escape or mixed), determined based on the ERQ criteria for circuits with different synaptic thresholds.

b) How is the mechanism changing? Can you state that explicitly? I interpret that the extremes of your HCO hybrid system as being escape and release with reality sitting in the middle, which is mixed. Are you postulating distinct mixed mechanisms? Can you define them?

We added a clarifying sentence to lines 309-312, stating that there is a continuum of mechanistic interactions in the hybrid half-center oscillators weighted differently by the mixtures of escape and release mechanisms, with the synaptic escape and release lying at the ends of the continuum. We also added two paragraphs to the Discussion section talking about the mixtures of mechanisms, ERQ measure and the difficulties associated with identifying exactly which mechanisms are involved (lines 722-743). The asymmetry in the intrinsic excitability and resting membrane potentials of the cells in a circuit adds complexity to a precise identification of the mechanism of oscillations, because asymmetry can result in two cells making on-off transitions with slightly different balances in the mechanistic interactions. To illustrate that this can be the case, we calculated the ERQ values for the two cells independently and showed the associated mechanisms of oscillation in the maps (Figure 4C,D).

c) I think that you are saying that the balance between release promoting ionic currents and escape promoting ionic currents are changing but you do not define these currents. In the mixed regime, one should speak about ionic currents if one is to speak of mechanism.

We added a few sentences to the Discussion section describing the ionic currents involved in the on-off transitions in the mixture of mechanism (lines 736-743). At the intermediate synaptic thresholds, on-off transitions are caused by a buildup of the H current in the inhibited cell, and a decay in the synaptic current. Decay in the synaptic current is associated with the hyperpolarization of the membrane potential of the active cell caused by the decay of the H current and an increase in the synaptic current of the inhibited neuron. The degree to which changes in the synaptic and H currents contribute to the transitions depend on the position of the synaptic threshold within the slow wave.

10. Lines 289-291: What about systems that rely on spike-mediated transmission?

This statement refers to the models with graded synapses. We specified this in the text (line 354). The presence of action potentials and/or spike-mediated transmission extends the range of synaptic thresholds over which oscillations can occur.

11. Line 294: In what sense is bursting less regular? Can you define the criteria and present numerical data? Maybe Coefficient of Variation of the cycle period or frequency?

As suggested, we calculated the Coefficient of Variation of the cycle frequency for all the synaptic thresholds and combinations of the synaptic and H conductances. These data are included as a Figure 4 —figure supplement 1A. Mean CV for the intermediate values of the synaptic thresholds (middle maps) is higher than for the extreme thresholds corresponding to the escape and release cases (left most and right most maps).

12. Lines 296-301: Can you provide evidence for these conclusions by analyzing each cell separately for ERQ? Are you saying that in mixed mode that one cell is escaping, and one is being released? What exactly are you thinking here?

As suggested, we calculated the ERQ for each cell individually and included these data in Figure 4C,D instead of the maps showing the mean ERQ calculated across two neurons. We also added a paragraph discussing the differences in ERQ values and mechanisms of transitions between the neurons (lines 365-374). In the case of the asymmetric oscillations the two cells make on-off transitions with slightly different balances in the mechanistic interactions: escape, when the synaptic threshold is close to the bottom of oscillation (typically associated with low synaptic conductance and high H conductance) and release, when the synaptic threshold is close to the top of the oscillations (typically associated with high synaptic conductance and low H conductance). Identifying exactly the mechanistic interactions requires knowing all the underlying currents.

We included paragraphs to the Discussion section talking about the ionic current involved in the transitions and talking about the challenges in identifying the exact mechanistic interactions in the mixed regime (lines 722-756).

13. Line 312: STAR methods?

Fixed

14: Lines 320-323: Asymmetry in activity does not require asymmetry in excitability. You could have bistability or a (possibly even artifactual) difference in baseline membrane potential despite your attempts to equalize them with injected current. This statement requires more evidence to be firm.

We meant that asymmetry in oscillations can result from asymmetry in intrinsic properties of neurons, including the differences in baseline membrane potentials. We fixed the phrasing in the text. We agree that bistability can be one of the reasons for asymmetry, however, we are confident that in our case asymmetry in oscillations resulted from the asymmetry in excitability and resting membrane potentials between the cells. In our experiments we repeated some of the same combinations of gH and gSyn, resulting in different initial conditions, always obtaining the same type of asymmetry with one neuron is dominating over the other. To further support our claim, we calculated the difference in the burst durations between the neurons for different combinations of synaptic and H conductances as a measure of asymmetry and included this analysis as Figure 4 —figure supplement 1B.

15. Lines 327-341: What values were chosen for gh and gSyn in each case (escape vs release)? Are these the same throughout all experiments???? How about Vth, is this the same across experiments for each case?

We added a paragraph to the methods section describing the maximal conductances for the synaptic and H currents and the values of the synaptic thresholds that were used across preparations and mechanisms (lines 1042-1052). We also discussed where the variability in the synaptic thresholds and conductance values comes from.

For the escape mechanism, the synaptic thresholds were between -54 and -52 mV. Small variability in the synaptic thresholds comes from the variability in the resting membrane potentials of the neurons across preparations (Figure 5C). For the release mechanism, synaptic thresholds were between -44 and -32 mV. Variability in the synaptic thresholds for release mechanism comes from both the differences in the resting membrane potentials and the intrinsic excitability of the cells, such as spike threshold (Figure 5D) number of spikes per burst and spike frequency, since, as you pointed out, in release, the transitions more heavily rely on spike-mediated transmission.

There was a larger variability in the maximal conductances used across temperature experiments due to a significant variability in the excitability of the cells comprising the circuits across preparations. gSyn varied between 200 and 900 nS and gH varied between 300 and 900 nS. More excitable neurons generated oscillations at lower values of the artificial conductances. Generally, the values of the maximal conductances were chosen as the lowest values resulting in stable oscillations at 10^o^C.

It is practically impossible to generate the same mechanism for the same values of conductances and synaptic thresholds across preparations due to biological variability. That is one of reasons we introduced the ERQ in this paper.

16. Line 344: Can you quantify this irregularity as CoV? I see an exemplar that seems more variable but also a lot faster (higher frequency). Are there pooled data to support claims on cycle frequency and its variability?

As suggested, we quantified the irregularity of the rhythms by calculating the CV of cycle frequency and spike frequency of the circuits in release and escape at low and high temperatures. The CV of cycle and spike frequencies of release, but not escape, circuits are significantly higher at higher temperatures. These data are added to the paper as a supplementary figure to Figure 5 and to the text (lines 422-425). For this analysis we included the circuits with a release mechanism that stopped oscillating before the temperature reached 20^o^C. Thus, in some cases, the properties of the circuit output were analyzed at temperatures below 20^o^C, when the circuits were still producing antiphase oscillations.

17. Lines 346-348: Is this surprising? The exemplar is dependent solely on spike-mediated transmission and so is susceptible to changes in spiking.

This is not surprising, significant reduction in the spike amplitude and hyperpolarization of the membrane potential of the neurons causes the circuits with release mechanism to be more susceptible to an increase in temperature, as discussed in the “Effect of temperature on the intrinsic properties of GM neurons” section.

18. Lines 385-388: This seems an understatement. This system at elevated Vth is operating in spike-mediated transmission mode and so spikes and spike frequency are all important.

We changed the phrasing to stress the role of spikes in switching behavior in the case of depolarized synaptic thresholds (line 471).

19. Lines 394-395: I don't see this in the data. I see a distinct crossover at around 4 nA – 15Hz. Can you quantify slopes? Is there pooled data to support this conclusion?

We calculated the slopes of all the f-I curves (N=13) at 10^o^C and 20^o^C and plotted the values on the unity line plot. We included the statistics for the slopes to the text (lines 479-480). For this analysis we used a near-linear portion of the f-I curves at low injected currents. We added a paragraph to the Methods section describing the slope calculations and the stats we performed (lines 1150-1153). In Author response image 3 we are also including an example of f-I curves at 10^o^C and 20^o^C from a different experiment.

**Author response image 3. sa2fig3:** 

20. Lines 402-411: You are here recognizing the distinct difference between HCOs that operate with graded vs spike-mediated inhibition. This should be thoroughly aired in relation to Figure 1; don't wait till here.

We added a paragraph describing graded vs spike-mediated transmission in escape and release modes in the beginning of the Results section (lines 173-179).

21. Lines 411-413: Can you support this conclusion with data?

Circuits with a release mechanism are sensitive to temperature increase, because it causes membrane potential hyperpolarization and reduction in spike amplitude. Thus, the degree to which temperature affects these characteristics will affect the degree of sensitivity of release circuits to temperature. In Figure 5C-F we show that there is a large variability in the intrinsic properties of GM neurons, as well as variability in sensitivity of these properties to temperature, as can be seen from the slopes of the lines in Figure 5C,D.

22. Lines 418-421: There are real difference between a slow wave HCO intrinsic excitability and a spiking HCO. For one a weakly regenerative current like a INaP can lead to weak rebound, but robust spiking supporting an HCO based on spikes. I suspect in the case Figure 5B1 that you either have a very long lingering h-current or a LT relatively slowly inactivating Ca current that is fully inactivated when Vth is more positive and inhibition weaker. You do not measure rebound in the GM neurons.

Artificial H current causes the rebound in GM neurons. If one of the neurons is much less excitable (has significantly lower input resistance) than the other neuron, H current might not produce enough depolarization in a less excitable cell to cross the synaptic threshold and escape the inhibition of a more excitable cell.

23. Figure 6: I find the logic of this figure unclear. The purpose is to study the effect of temperature on the hybrid HCOs in different oscillation modes. So, you present HCOs with the added h and synaptic conductances at standard temperature for a comparison group (Panel A), but then Panel B is simply to illustrate changes associated with changes in h and synaptic conductances at a different temperature, and finally in Panel C you explore a simulated change in temperature. I understand why you have Panel B included (to parse mechanism) but shouldn't the order be ACB?

The logic is as follows – we introduce temperature dependence in the computer-generated currents one by one to parse the mechanism of temperature induced changes in the circuit output. First, temperature-independent currents (panel A), second, we implemented the temperature dependence in the conductances only (panel B), third, we implemented the temperature dependence in both conductances and activation rates to see how changes in the activation rates change the frequency responses of the circuits (panel C).

24. Lines 451-453: Do you have data to support this conclusion?

In a few experiments, neurons with substantially different excitability properties produced asymmetric oscillations at low temperature. In case of temperature-dependent artificial conductances, the asymmetry was amplified as temperature was increased, resulting in the cessation of oscillations at high temperatures. However, we do not have enough data to make a strong conclusion regarding unstable oscillations, thus, we removed this sentence from the text.

25. Line 456: '…temperature-independent synapse…' This is getting confusing. It is temperature insensitive h-current and synapses; please make it clear.

Fixed

26. Line 460: '…release or escape…'

Fixed

27. Lines 479-480: Here it would be good to remind the reader that the deactivation rate is as important as the activation rate and that in fact they are two sides of the same coin. From Methods I think you just changed Tau0s (2X or 1.5X) (τH0 2000 or 3000 msec, τ*s*yn 50 or 100 msec) with temperature so both activation and deactivation are affected. I suggest that you plot the Taus of the synapses and the h-current on the plots of Figure 1B. You do provide the equations in Methods, but I think a visualization would help.

Activation (deactivation) rate of the synaptic current does not depend on the membrane potential. Activation (deactivation) rate of H current does not significantly change in the operational range of membrane potentials of half-centers in our study, thus, we decided to omit the activation rates plot from Figure 1. We are including the plot in Author response image 4 for your reference.

**Author response image 4. sa2fig4:** 

28. Lines 486-489: Any data to support these conclusions? Change activations rates without affecting deactivation?

This conclusion follows from combining three observations: (1) increase in the oscillation frequency in the case of temperature-independent synapses and H current (Figures 6A, 7A,E); (2) decrease in the oscillation frequency in the case of temperature-dependence in the conductance of the synaptic and H currents (Figures 6B, 7A,E); (3) increase in the oscillation frequency with the decrease in activation (deactivation) rates of the synaptic and H currents (Sharp et al., 1996, Figure 7; data from our experiments). Increase in the oscillation frequency in the first case results from the changes in the intrinsic properties of the neurons, illustrated in Figure 5 – hyperpolarization of the membrane potential and a reduction in the spike amplitude, causing a decrease in synaptic and, consequently, H current. Decrease in the oscillation frequency in the second case with the increase in synaptic and H conductance is shown and discussed in Figure 3. Acting together, these changes counteract each other, and can result in a nearly constant oscillation frequency when the temperature is increased.

29. Lines 491-492: Can you provide more evidence in support here? I would like to know what is causing the rebound in the GM neurons before I can fully accept this conclusion. If the rebound is due to lingering h, maybe emphasize the deactivation.

The rebound is caused by the activation of H current. We added a sentence clarifying it (line 580).

30. Lines 518-521: Please put the numbers here in the text, at least for the Q10 g and k case. The Table is pretty tough to isolate on.

We put the ERQ values for release and escape circuits at 10^o^C and 20^o^C and p-values for all the Q10 cases in the text (lines 609-617).

31. Lines 521-525, Figure S2: This exemplar in Figure S2 is not convincingly escape at 10 C. Pure escape should transition to the depolarized state when threshold is reached. This is a good place for me to comment on the ERQ criterion for designating escape and release. I appreciate the need for an automatable algorithm to designate escape vs. release but I have two caveats. (1) Classically escape is designated by transition when the inhibited cell reaches synaptic threshold (or maybe in spike-mediated transmission spiking threshold) and release is designated by transition when the depolarized cell crosses the synaptic threshold (or stops spiking). The case in Figure S2 clearly violates this classical designation. (2) ERQ is determined by averaging across the two neurons in the HCO, this hides asymmetries and allows for the two cells to have different mechanism, e.g., one escape and one mixed. ERQs must be calculated independently for the two neurons if you wish to reveal asymmetries.

You are right, the case in Figure S2 (now Figure 7 —figure supplement 1) it is not a pure synaptic escape mechanism at 10^o^C, the mechanism is likely an intrinsic escape mechanism or a mixture of the intrinsic escape and synaptic release, because the synaptic threshold is closer to the bottom of the slow wave, but the transition does not happen at the synaptic threshold as it would in the case of synaptic escape and there is a decay in the synaptic current allowing the neuron to be released from inhibition.

Further, as you pointed out, the neurons shown in the figure are different with respect to their membrane potentials and excitability properties, resulting in the on-off transitions in these neurons being governed by different balances of mechanistic interactions at lower temperature. We calculated the ERQ for each neuron independently to uncover the differences in the mechanisms between the cells and updated Figure 7 —figure supplement 1. We altered the text to describe the mechanism of oscillation more accurately based on the new ERQ analysis (lines 617-620). The new analysis, however, did not alter the main message we were trying to convey with this figure, illustrating that temperature can alter the mechanism of oscillation. The ERQ of both neurons changed significantly when the temperature was increased.

We also added paragraphs to the Discussion section describing the caveats of the ERQ measure and the difficulties associated with identifying exactly which mechanisms are involved in oscillation generation when neurons are asymmetric (lines 722-743).

32. Lines 547-550: Only one prep is illustrated in Figure 8C. Is the CoV of bursting in mixed mode different across pooled data to support this statement?

We calculated CV of cycle frequency of all the circuits in escape, mixture and release mechanisms in control and with addition of I_MI_. We included these data in Figure 8 —figure supplement 1. The mechanism of oscillation in each case was determined based on the ERQ thresholds calculated using the maximum and minimum of the second derivative of the sigmoid functions that were fit to ERQ vs V_th_ data, as described in lines 223-226. There was a significant increase in the CV of cycle frequency of the circuits operating in mixed mode with I_MI_ relative to control. We included the stats in the text (lines 644-646). Here, we are also including several more examples of the voltage traces showing and increase in irregularity of bursting and asymmetry between the cells with the addition of I_MI_.

33. Line 590: '…vastly…'

We deleted “vasty”

34. Lines 593-597: Only one prep is illustrated in Figure 8E. Are there pooled data that can support this conclusion?

We performed 4 experiments, in which oscillations of the circuits were lost at high temperature and restored by adding I_MI_ to both neurons. We specified in the text of the manuscript that the oscillations were restored in 4/4 circuits with the addition of I_MI_ (line 699). We have provided a single example trace in the paper, because the effect of I_MI_ was consistent across all the preparations. Here, we provide more examples of I_MI_ rescue. Red bars indicate when I_MI_ was added.

35. Lines 650-653: Because asymmetry is not directly assessed in this paper this conclusion should be scaled back.

In new analyses we assessed the asymmetry by calculating the difference in the burst durations between the two neurons in the circuit in different modes and for different combinations of synaptic and H conductances (Figure 4 —figure supplement 1B). We also calculated the ERQ values for the two cells independently to show the differences in the on-off transitions between the neurons (Figure 4C,D). Further, we calculated the differences in the number of spikes per burst between the neurons in circuits in control and with addition of I_MI_ (Figure 8 —figure supplement 1B). In the cases when the neurons had similar number of spikes per burst in control conditions, addition of I_MI_ did not destabilize the circuits with the mixture of mechanisms. In the cases when the neurons had substantially different numbers of spikes in control, I_MI_ amplified this difference (lines 650-653). Based on this finding, we conclude that not just the mechanisms of oscillations but also the degree of asymmetry between the units influences the way reciprocally inhibitory circuits will respond to modulation.

36. Lines 653-656: Only one prep is assessed in mixed mode, so this statement should be scaled back.

In the new analysis we assessed the irregularity and asymmetry between the cells in all the experiments in the mixture of mechanisms to support this conclusion (Figure 8 —figure supplement 1).

37. Lines 687-689: Please be careful here and designate modes precisely and state exactly what you mean by stable cycle frequency. Does this mean regularity of bursting or period constancy? Ditto for living preparations with intact networks.

Synaptic and H-conductances work against each other in escape circuits, maintaining the oscillation period constant throughout a wide range of temperatures. This observation is also illustrated in Figure 3B. We changed the phrasing of this sentence in the text clarifying the conditions when this holds true (lines 856-858).

Line 693: does temperature compensation in intact STG networks involve a constant period of just constant phase?

Temperature compensation in STG networks involve constant phase but not constant period (Tang et al., 2010, Powell et al., 2021). Cycle frequency of both pyloric and gastric STG circuits increases with temperature.

38. Lines 626-628: Are there modeling studies involving more realistic neurons specifically ones that spike and use spike-mediated transmission.

Thank you for reminding us of the modeling studies describing the systems that rely on both graded and spike-mediated transmission and contain elements of both escape and release modes. We discuss these models and regulation of oscillations in these models by parameter variations in lines 779-791.

Reviewer #2:This manuscript provides a very detailed and thorough examination of an important issue in neural circuit research, namely how the mechanisms underlying neural activity relate to robustness in the face of perturbations. It examines the simplest neural circuit possible, one involving just two neurons that reciprocally inhibit each other, which is capable of producing rhythmic alternating activity. The research shows that there is a continuum of mechanisms based on synaptic and membrane properties of the two neurons that can generate a robust output. At one end of the continuum, each neuron escapes from the inhibition of the other. At the other end, each neuron releases the other from inhibition. In the middle, both mechanisms contribute to generation of rhythmic activity. The effects that perturbations such as temperature and neuromodulators have on the circuit depend upon where the mechanism of oscillation lies along this continuum.This paper has several important strengths:It uses dynamic clamp technique to artificially couple two real neurons and provide them with a membrane conductance that they don't normally have. This is a powerful technique that merges experimental and theoretical neuroscience because the researchers are able to systematically alter parameter values such as synaptic strength and ionic conductance that are not feasible to modify biologically. Yet they are also monitoring the activity of real neurons.The manuscript thoroughly represents the results and convincingly demonstrates how release and escape mechanisms are differentially affected by perturbations. The method of data visualization is very effect at summarizing complex results.An important conclusion drawn from the results is that half-center oscillators using a release mechanism are more robust to variations in synaptic and membrane conductance.Another important conclusion is that the same circuit can produce a similar output using different mechanisms and that it is not possible to know which mechanism is used without looking at the effect of perturbation.I would encourage the authors to not start the abstract with a question, but rather use a standard topic sentence that gets right to the problem.

We rewrote the abstract and removed the question.

The introduction could be firmed up more. For example, the sentence "Lateral inhibition is important in many sensory systems, and reciprocal inhibition between individual neurons or groups of neurons is the 'building block' of many half-center oscillators that generate antiphase and multiphase activity patterns." Is a run-on referring to lateral inhibition in sensory systems and then going into detail about reciprocal inhibition in HCOs. The concluding sentence of the introductory paragraph does not follow from the content of the paragraph.

We rewrote and reorganized much of the Introduction to have a better flow of ideas and to address the review questions, such as defining circuit robustness.

Figure 3A. If I understand this correctly, 10 models were made with each of the 49 combinations of gH and gSyn. The percentages then are going to limited to 0,10,20,30,40…100%. To make that clear, change the continuous gray scale "%oscillators" to the 10 discrete gray values (as was done in Figure S1A). That will provide the reader with more information about the values. Same with Figure 3F.

We changed the continuous gray scale "%oscillators" to the 10 discrete gray values.

Figure 3C, the red boxes are nearly invisible and I imagine that Figure 3D is not color-blind friendly.

We changed the color of the red boxes to black and increased the line thickness for better visibility. We used a color blind simulator to transform the figures to simulate different types of colorblindness, and the color gradient is visible in different color blindness conditions.

Figure S1A, I found the colors difficult to distinguish. The trends did not pop out at me.

We changed the brightness of the colors in Figure S1D (now Figure 4—figure supplement 1F) to be more distinguishable.

Figure 7 refers to 'cases' that are also in Figure 6, but not referred to as cases. It took me a while to recognize that they were the same. It would be helpful to label case 1, case 2, and case 3 in Figure 6 A1, B1, C1 and in Figure 7 A-D in the Figure It would also help to refer the cases in the text for consistency such as line 438, Case 2 and line 475 Case 3.

As suggested, we labeled case1, 2 and 3 in Figure 6 and Figure 7 A-D. We also referred to the cases 1, 2 and 3 in the text (lines 419, 511, 512, 520, 524, 561, 610-617).

Also, recognizing color equivalence in Figure 7 A-F lines and E-H boxes is really hard. I think it's because the box plots have a contrast-enhancing black border. I'm starting to think that I may be color-impaired.

Thank you for pointing out this inconsistency. The opacity of the colors in panels A-D and E-H was different, making the colors look different. We fixed the colors and removed the black boarders in bar plots so that the color equivalence is easier to recognize.

Line 546-550: "I-MI made oscillations less stable and irregular for circuits operating with a mixture of mechanisms." Stating "This is obvious…" is not an explanation; although it may be obvious to the authors, it needs to be explained to the readers.Regarding the 2nd point, "an increase in the standard deviation of the cycle frequency", I don't see documentation of this; the error bars in Figure 8C are larger at the release end of the graph than in the middle.

We meant that I_MI_ destabilizes the circuit by amplifying the asymmetry between the units. We changed the phrasing in the text (lines 646-653). We calculated the CV of cycle frequency of circuits in control and with addition of I_MI_ and added these data to Figure 8 —figure supplement 1. There was a significant increase in the CV of cycle frequency of the circuits operating in mixed mode with I_MI_ relative to control. We included the stats in the text (lines 644-646). To assess asymmetry, we calculated the difference in the number of spikes per burst between the neurons in half-centers in control and with addition of I_MI_ (Figure 8 —figure supplement 1B). 3/8 preparations did not oscillate at intermediate synaptic thresholds when I_MI_ was added, exhibiting the extreme case of asymmetry when one of the neurons was suppressing the other one the whole time. In 3/5 circuits that remained bursting the was a higher different in the number of spikes per burst between the neurons in circuits with a mixture of mechanisms (Figure 8 —figure supplement). Overall, the effect of I_MI_ on irregularity and asymmetry of oscillations is very variable and depends on many factors, including how symmetric the oscillations are in control condition.